# Tunneling current-controlled spin states in few-layer van der Waals magnets

ZhuangEn Fu [1,2], Piumi I. Samarawickrama[1,2], John Ackerman[3], Yanglin Zhu[4], Zhiqiang Mao [4], Kenji Watanabe [5], Takashi Taniguchi [6], Wenyong Wang[1,2], Yuri Dahnovsky[1,2], Mingzhong Wu[7], TeYu Chien [1,2], Jinke Tang[1,2], Allan H. MacDonald [8], Hua Chen [9] ✉ & Jifa Tian[1,2] ✉

Effective control of magnetic phases in two-dimensional magnets would constitute crucial progress in spintronics, holding great potential for future computing technologies. Here, we report a new approach of leveraging tunneling current as a tool for controlling spin states in CrI$_3$. We reveal that a tunneling current can deterministically switch between spin-parallel and spin-antiparallel states in few-layer CrI$_3$, depending on the polarity and amplitude of the current. We propose a mechanism involving nonequilibrium spin accumulation in the graphene electrodes in contact with the CrI$_3$ layers. We further demonstrate tunneling current-tunable stochastic switching between multiple spin states of the CrI$_3$ tunnel devices, which goes beyond conventional bistable stochastic magnetic tunnel junctions and has not been documented in two-dimensional magnets. Our findings not only address the existing knowledge gap concerning the influence of tunneling currents in controlling the magnetism in two-dimensional magnets, but also unlock possibilities for energy-efficient probabilistic and neuromorphic computing.

The rise of spintronics as an important axis in modern computing technology has been primarily anchored by the utilization of spin states in magnetic materials. Magnetic tunnel junctions (MTJs)[1], consisting of two ferromagnetic layers separated by an insulating barrier, are among the most important components for spintronics applications. The control of spin states in MTJs, primarily via spin-transfer torque (STT)[2] and spin-orbit torque (SOT)[3], has proven invaluable in a wide range of spintronic devices[1], encompassing magnetic random access memory (MRAM)[4,5], hard disk drive read heads[6], radio-frequency sensors[7], and even artificial probabilistic and neuromorphic computing[8–10]. As the imperatives of device miniaturization and energy efficiency become paramount, conventional MTJs

encounter major challenges in precise control over the thickness of constituent layers, in ensuring high-quality interfaces between ferromagnetic and barrier layers, and in accommodating the large current densities (on the order of $10^6$ A/cm$^2$ or higher) needed for magnetization switching[5,11,12]. These obstacles call for the exploration of new materials, physical principles, and architectures that can be tailored to fulfill the rigorous demands of next-generation computing devices.

A new paradigm has recently been introduced through two-dimensional (2D) van der Waals (vdW) magnets[13–22]. In contrast to ultrathin conventional ferromagnetic metals, 2D vdW magnets often have substantial intrinsic magnetocrystalline anisotropy even down to their monolayers. In particular, the wide flexibility of 2D vdW magnets

[1]Department of Physics and Astronomy, University of Wyoming, Laramie, WY 82071, USA. [2]Center for Quantum Information Science and Engineering, University of Wyoming, Laramie, WY 82071, USA. [3]Department of Chemical Biomedical Engineering, University of Wyoming, Laramie, WY 82071, USA. [4]Department of Physics, The Pennsylvania State University, University Park, PA 16801, USA. [5]Research Center for Electronic and Optical Materials, National Institute for Materials Science, 1-1 Namiki, Tsukuba 305-0044, Japan. [6]Research Center for Materials Nanoarchitectonics, National Institute for Materials Science, 1-1 Namiki, Tsukuba 305-0044, Japan. [7]Department of Physics and Department of Electrical and Computer Engineering, Northeastern University, Boston, MA 02115, USA. [8]Department of Physics, The University of Texas at Austin, Austin, TX 78712, USA. [9]Department of Physics and School of Advanced Materials Discovery, Colorado State University, Fort Collins, CO 80523, USA. ✉e-mail: huachen@colostate.edu; jtian@uwyo.edu

with different types of magnetic anisotropy and exchange interactions suggests easy engineering of the magnetic phases, which is crucial for reducing or strengthening spin fluctuations or inducing various forms of magnetic order. For instance, $CrI_3$ exhibits ferromagnetic ordering within monolayers and weak antiferromagnetic ordering between adjacent layers[14]. In contrast to conventional MTJs[23–25], tunnel junction devices based on few-layer $CrI_3$ have several distinct characteristics: (1) the magnetic layers concurrently act as tunnel barriers and are insulating; (2) the transport process through the magnetic layers is expected to be coherent rather than incoherent as in ferromagnetic metals; and (3) the magnetic moments in the magnetic layers are always ideally collinear in both the ground and metastable magnetic configurations under finite magnetic fields[14,26]. Over recent years, there have been successful efforts in manipulating magnetic properties[14,16,17] in 2D vdW magnets through various means such as electrostatic doping[18,19], external pressure or strain[27,28], and illumination at specific light wavelengths[29]. Despite these advances, the spin-polarized tunneling current has primarily been utilized as a means for detecting magnetic states, while its critical role in controlling spin states in few-layer 2D vdW magnets, especially among the corresponding magnetic domains, remains elusive. Furthermore, a captivating and largely uncharted area of research is the potential to achieve both stable and metastable spin states within a single MTJ. The integration of atomically thin 2D vdW magnets, which possess distinct properties such as sharp interfaces, a natural and adjustable vdW insulating gap, and layer-by-layer control of thickness, could be the key to unlocking this possibility.

Here, we report the realization of a tunneling current-controlled unidirectional transition between two spin states in few-layer $CrI_3$ tunnel junction devices. Specifically, we demonstrate, at relatively low tunneling currents, an unusual, current-induced unidirectional spin-state transition between the spin-parallel (SP) and spin-antiparallel (SAP) states with the switching direction determined by the polarity and amplitude of the bias. We interpret this observation in terms of the nonequilibrium spin accumulation in the graphene electrodes in contact with the $CrI_3$ layers due to the spin-polarized tunneling current, and illustrate the mechanism using a Keldysh nonequilibrium Green's function (NEGF) method. Furthermore, at slightly higher biases (but still in the few nA and μA range), we demonstrate stochastic switching between two or three metastable spin states in few-layer $CrI_3$, with the number of states being contingent on the number of layers. Remarkably, the $CrI_3$ tunnel junction devices show exceptional energy efficiency, with power consumption about three orders of magnitude lower than that of traditional MTJs. Our work offers seminal insights into the role of tunneling current in modulating spin states in few-layer 2D vdW magnets, representing a significant leap forward in studying 2D magnetism and paving a potential way for developing highly energy-efficient, next-generation computing technology.

## Results and discussion

### Transport signatures of magnetic domains in few-layer $CrI_3$

We have fabricated high-quality tunnel junction devices composed of few-layer $CrI_3$ using the vdW assembly method[30–32] (see Methods). Figure 1a exhibits a schematic (top) and optical image (bottom) of a tunnel junction device with a bilayer (2L) of $CrI_3$ (~1.2 nm). Figure 1b shows the relationship between temperature ($T$) and tunneling resistance measured at zero magnetic field, as observed in the graphite/$CrI_3$(2L)/graphite tunnel junction device. At $T$ ~43 K, we see an apparent resistance kink which corresponds to a magnetic phase transition from paramagnetic ($T > 43$ K) to antiferromagnetic (AFM) ($T < 43$ K) ordering. Importantly, the extracted Néel temperature ($T_N$) of about 43 K matches with previously reported results using alternate methodologies[14,33]. Furthermore, as shown in Fig. 1c, 2L $CrI_3$ operates as a magnetic tunnel barrier within the tunnel junction device.

We next measure the tunneling resistance as a function of the applied magnetic field perpendicular to the sample plane of the 2L $CrI_3$ tunnel junction device at $T = 1.5$ K, as shown in Fig. 1d. We see that the magnetoresistance shows two predominant resistance states. At magnetic fields near zero, there is a high resistance state around 9.5 MΩ, which corresponds to the layer-AFM states (↑↓ or ↓↑), which represent the ground state of the 2L $CrI_3$. Conversely, at higher magnetic fields, a low resistance state of ~3 MΩ emerges, signifying the layer-FM states (↑↑ or ↓↓). Strikingly, we observe that the substantial change in resistance from 3 to 9.5 MΩ is comprised of several smaller, step-like changes, as shown in Fig. 1d (also see Supplementary Fig. 1a–c). This implies the presence of intermediate magnetic states during the layer-magnetic phase transition. The step-like change in tunneling resistance is expected due to different spin configurations of the magnetic domains in the adjacent $CrI_3$ layers, as illustrated in Fig. 1e. We note that a similar phenomenon has been previously observed[16,17,34] and has ascribed to the existence of magnetic domains but lacked further characterization.

We further systematically study the tunneling resistance vs. magnetic field at different cooling histories (Supplementary Fig. 1a–c). We find that coercivities and magnitudes of the mini step-like resistance changes exhibit a high dependence on the cooling history, which is consistent with features of magnetic domains. In stark contrast, the major step-like resistance changes, which are associated with the layer-magnetic phase transition, remain relatively invariant with respect to cooling history. Additionally, by examining the temperature dependence of the magnetoresistance (Supplementary Fig. 2), we observe that the mini step-like resistance changes in the 2L $CrI_3$ persist up to approximately 30 K. Remarkably, we discern that the coercivities correlated with the spin-state transitions, characterized by the step-like resistance or voltage changes, in the 2L $CrI_3$ can be tuned by the applied bias voltage/current (Supplementary Fig. 3). Specifically, we observe that the coercivity at the SAP to SP transition on the positive magnetic field side increases with the increasing bias voltage, whereas an opposite trend is observed for the same transition on the negative magnetic field side. We note that mini step-like resistance changes in the four-layer (4L) $CrI_3$ tunnel junction devices have also been revealed (Supplementary Figs. 1d–f, 2, 4). However, in contrast to 2L $CrI_3$, the 4L $CrI_3$ tunnel junction device exhibits a decrease in coercivity associated with the spin-state transition between ↑↑↑↓ (or ↑↑↓↑) and ↑↑↑↑ of (Supplementary Fig. 4) as the positive bias voltage increases, signifying a pronounced influence of the applied positive bias voltage on the magnetic properties of few-layer $CrI_3$. We note that while electrostatic gating[18,19] has a strong effect on the magnetism of 2D vdW magnets, its impact on our devices should be minimal compared to the spin-polarized current, which will be discussed later. Furthermore, the presence of magnetic domains in different 2D magnets, including few-layer $CrI_3$[35] has been confirmed through other techniques, such as single-spin microscopy and reflectance magnetic circular dichroism[35–38].

### Tunneling current-induced unidirectional spin-state transition

We now explore the tunneling current dependence on the spin-state transition in a few-layer $CrI_3$ near the layer magnetization transition region. Figure 2a depicts the voltage ($V$) as a function of bias current ($I$) for the 2L $CrI_3$ tunnel junction device under conditions of $T = 1.5$ K and a magnetic field of 0.55 T. The $V$-$I$ curve exhibits three prominent features: (i) hysteresis loops observable in the positive and negative current regions (see right and middle insets of Fig. 2a); (ii) voltage fluctuations around a bias current of ~4.5 μA (left inset of Fig. 2a); and (iii) conspicuous asymmetry in the $V$-$I$ characteristics. To elucidate the nature of the hysteresis $V$-$I$ loop observed in Fig. 2a, we probe the spin configurations preceding and following a rapid voltage transition, as indicated by the circled numerals 1 (at 1 μA) and 2 (at 2.2 μA) in Fig. 2a. Firstly, with the magnetic field fixed at 0.55 T, we sweep the current to

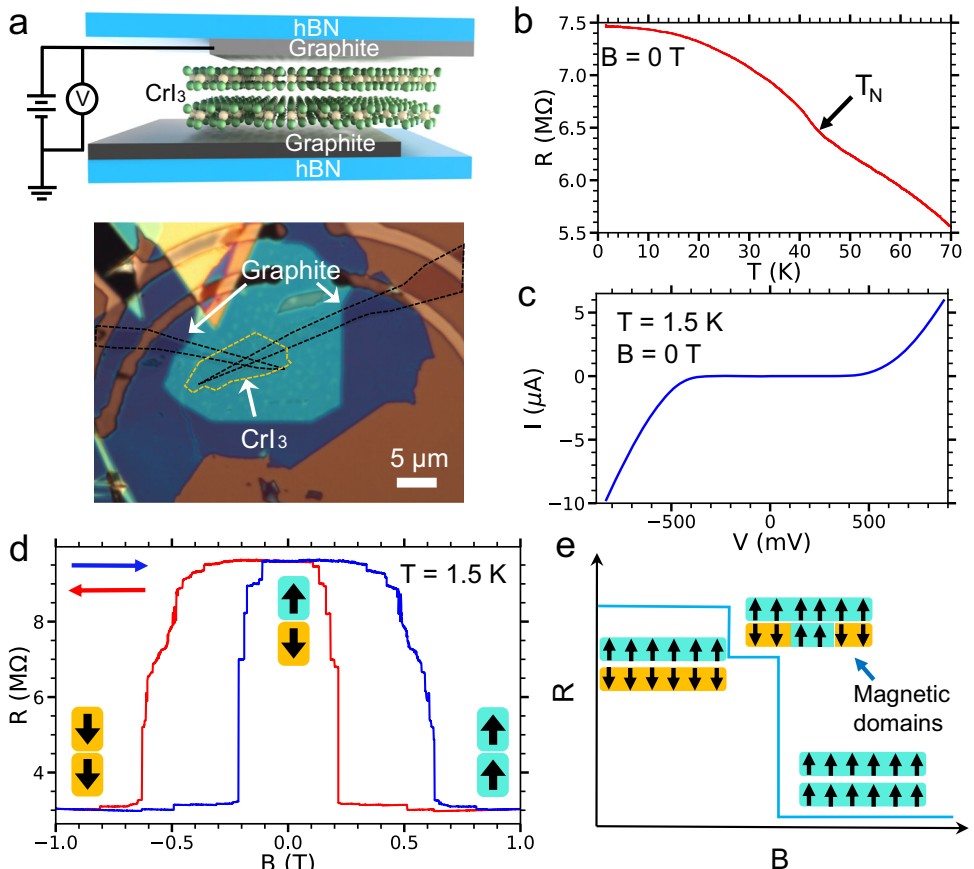

**Fig. 1 | Tunneling magnetoresistance and magnetic domain of a bilayer (2 L) CrI₃. a** Schematic of an hBN/graphite/CrI₃(2L)/graphite/hBN tunnel junction device (top panel). Optical microscope image of a 2 L CrI₃ tunnel junction device with two graphite layers as contacts for tunneling magnetoresistance measurements (bottom panel). **b** Temperature dependence of tunneling resistance ($R$) of 2L CrI₃ measured with a constant bias current ($I$) of 50 nA. **c** Tunneling current ($I$) as a function of applied bias voltage ($V$) at $B = 0$ T and $T = 1.5$ K. **d** Tunneling resistance as a function of applied out-of-plane magnetic field $B$ at a fixed bias voltage of 410 mV and $T = 1.5$ K. The Blue and red arrows indicate the magnetic field sweep directions. The black arrows in the insets indicate the out-of-plane magnetizations in the top and bottom CrI₃ layers. **e** Illustration of the magnetic-domain dependent resistance change.

5 μA before reverting it to the target currents of 1 μA and 2.2 μA, respectively. Subsequently, at these target currents, we ramp the magnetic field from its initial value of 0.55 T, peaking at 0.9 T, and then descending to 0.1 T. The corresponding results are presented in Fig. 2b. Remarkably, the difference between the magnetic states at these points is concomitant with a subtle, step-like resistance change, indicative of a transition from an SAP (↑↓ or ↓↑, circled 1) to an SP (↑↑ or ↓↓, circled 2) configuration. Consequently, the hysteresis *V-I* loops displayed in Fig. 2a at different bias currents are demonstrative of current-induced spin-state transitions among different magnetic domains in the 2L CrI₃. We further estimate the switching current density of the SAP to SP transition between the circled numerals 1 (at 1 μA) and 2 (at 2.2 μA), as shown in Fig. 2a, to be ~724 A/cm² (see Supplementary Note and Supplementary Fig. 5), which is notably three orders of magnitude lower than that reported values in previous studies employing SOT[39].

We further conduct an in-depth analysis of the current-controlled spin-state transitions under varying magnetic fields (Fig. 2c and Supplementary Figs. 6, 7), while remaining within the primary layer-magnetic phase transition region. This investigation reveals two salient phenomena: (i) the hysteresis *V-I* loop, attributed to the current-induced spin-state transition, migrates from the positive to the negative current region in response to increasing magnetic fields; and (ii) as the amplitude of the tunneling currents increases, the transition direction between SAP and SP is determined by the polarity of the current. For instance, the transition from the SAP to SP configuration,

corresponding to a high-to-low voltage state, is invariably detected at positive bias currents. In contrast, the transition from the SP to SAP configuration, characterized by a low-to-high-voltage state, is manifested at negative bias currents. It is important to note that we do not consider or reference the history of current sweeping. Furthermore, the tunneling current-induced unidirectional spin-state transition often occurs in relatively low-bias current regions. Consequently, our observation represents the first demonstration of tunneling current-induced unidirectional spin-state transition in 2D vdW magnets.

## Tunneling current-driven stochastic switching in 2L CrI₃

Another remarkable finding of this study is the observation of stochastic switching between metastable spin states in the graphite/CrI₃/graphite tunnel junctions. In the inset (left) of Fig. 2a, the representative results illustrate current-driven stochastic switching of a 2L CrI₃ device, where the voltage exhibits random fluctuations between two distinct values, when the bias current is modulated within the range from −4.2 to −4.8 μA. Subsequently, we conduct time-resolved measurements of the voltage fluctuations at a fixed current, maintaining the temperature at 1.5 K (see Fig. 2d and Supplementary Fig. 8). These measurements reveal that the voltage undergoes stochastic switching between two levels, corresponding to the SAP and SP states within magnetic domains of 2L CrI₃. The dwell time for each magnetic state is typically of the order of 10 ms, while the switching time between the states is around tens of microseconds, as depicted in the bottom panel of Fig. 2d. Figure 2e provides a schematic representation of the

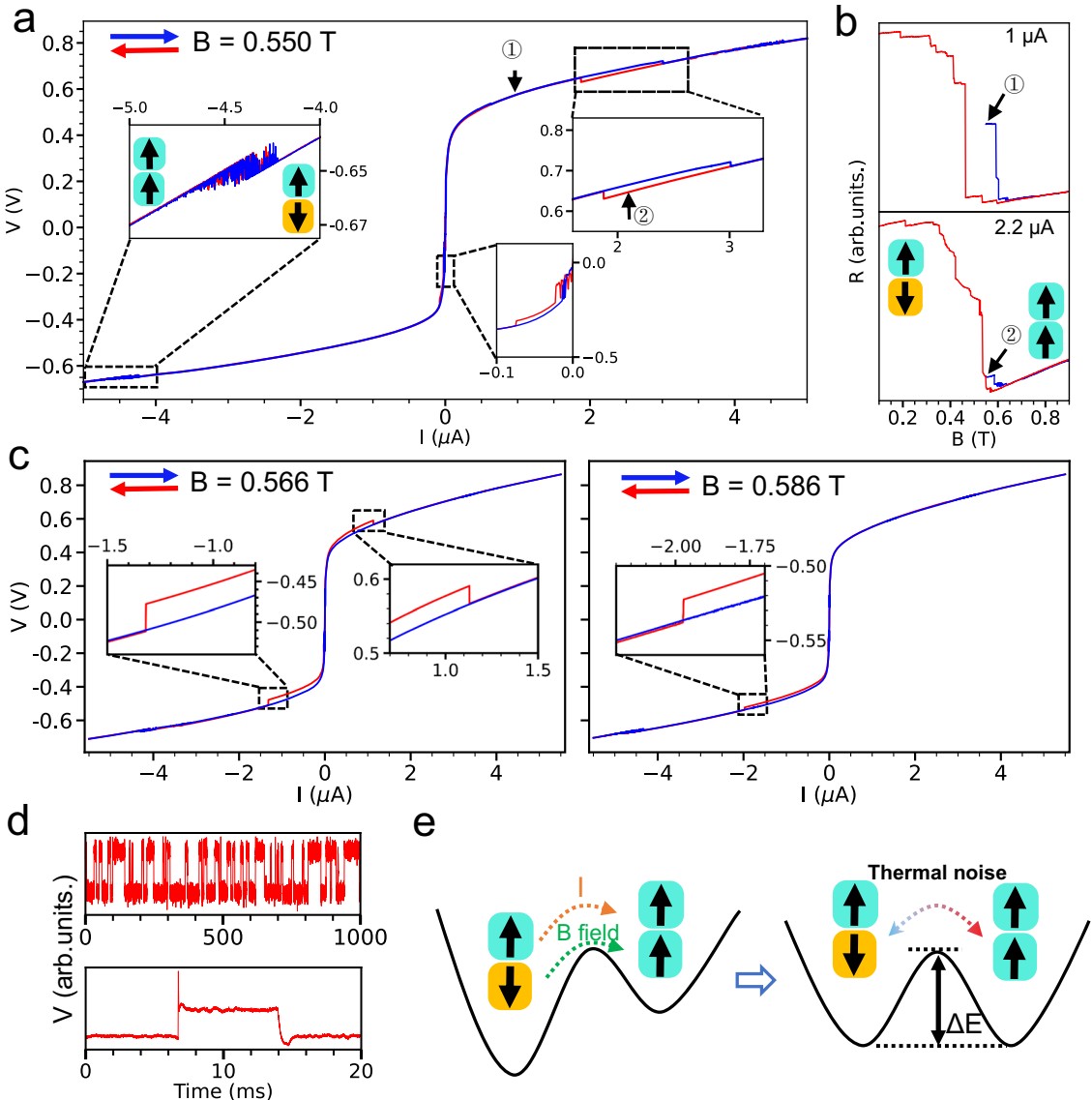

**Fig. 2 | Unidirectional current-driven magnetization reversal and stochastic switching of a 2L CrI₃. a** Voltage as a function of applied current of the 2L CrI₃ tunnel junction device at $B = 0.550$ T. The arrows inside the left zoomed inset indicate the spin configurations of magnetic domains. **b** Tunneling resistance as a function of the magnetic field at two bias currents of 1 μA (top panel) and 2.2 μA (bottom panel). The circled 1 and 2 in (**a**, **b**) indicate the corresponding initial magnetic states. **c** Voltage as a function of applied current of the 2 L CrI₃ tunnel junction device at $B = 0.566$ T (left panel) and 0.586 T (right panel). **d** Time snapshots of voltage for an applied DC current in the fluctuation region with time scales of 1 s (top panel) and 20 ms (bottom panel). **e** Schematic of magnetic domain-based stochastic spin-state switching between SAP and SP. All measurements were performed at $T = 1.5$ K. Zoomed insets are from the dashed black squares.

physical mechanism underlying the observed stochastic switching. It is established that the SP state is energetically disfavored relative to the SAP state, given that the ground state of the 2L CrI₃ is the SAP state. However, by applying an external magnetic field perpendicular to the plane, it is possible to effectively lower the energy of the SP state to match that of the SAP state. Due to the relatively small energy barrier ΔE near the layer-magnetic phase transition region, and in conjunction with thermal noise (caused by temperature and/or tunneling current-induced Joule heating) and spin accumulation (elaborated below), the system exhibits stochastic switching between the SAP and SP states within the magnetic domains under consideration.

We then investigate the dependence of stochastic switching on the tunneling current. Figure 3a shows the voltage as a function of applied bias current in a 2L CrI₃ tunnel junction device, measured at a magnetic field of 0.578 T and a temperature of 30 K. Notably, stochastic switching is observed for both positive and negative current regimes at this field. Furthermore, the tunneling current necessary to

induce stochastic switching is reduced by a factor of 20 compared to measurements taken at 1.5 K, highlighting the significant influence of temperature on the switching behavior (Supplementary Figs. 6, 9–11). A more detailed analysis of the tunneling current's effect on stochastic switching is provided in Fig. 3b, which comprises time snapshots of voltage measurements (left panels) at varying bias currents, accompanied by their respective voltage distribution histograms (right panels). We see that the tunneling current precisely controls the probability of the stochastic switching between low and high-voltage states. We have extracted the corresponding switching probabilities corresponding to high and low voltage states and summarized them in Fig. 3c. It is clear that as the bias current is increased from 100 to 300 nA, there is a transition from the SAP state to the SP state via a region of stochastic switching. In the present study, the CrI₃-based tunnel junction devices demonstrate remarkable energy efficiency in controlling stochastic switching compared to traditional techniques[40–44], which often consume much higher power

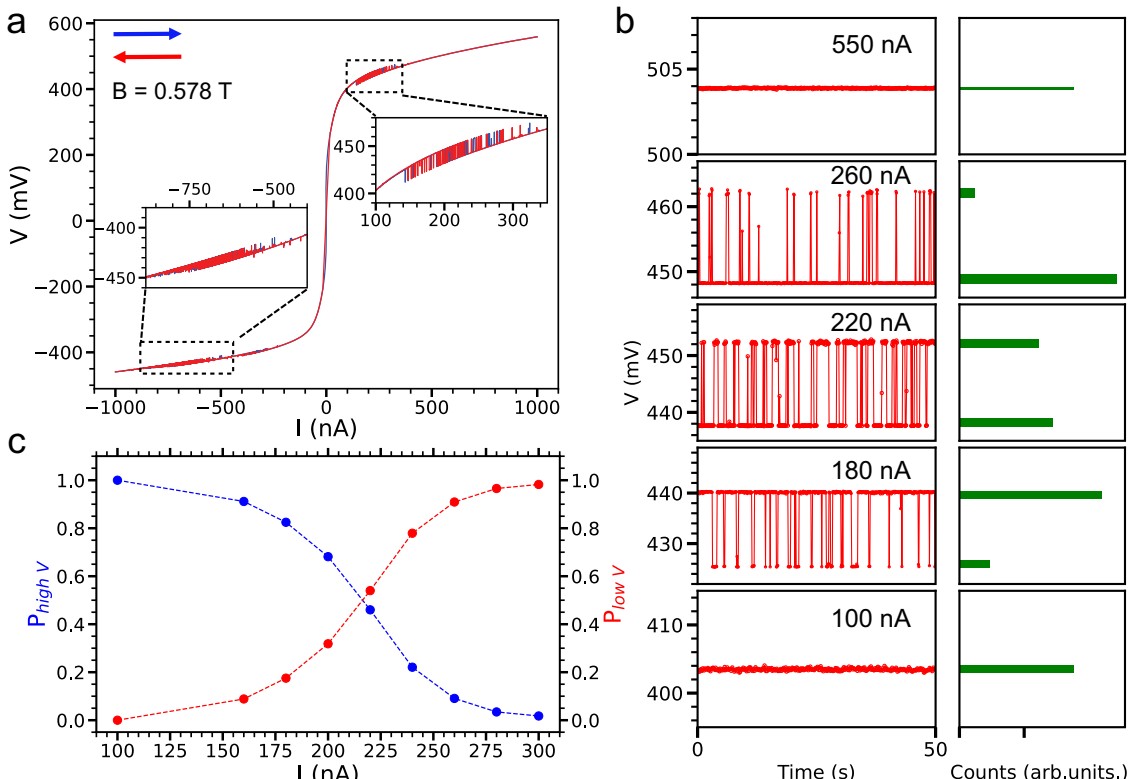

**Fig. 3 | Current modulated stochastic switching of a 2L CrI₃ at T = 30 K. a** Voltage as a function of applied current at $B = 0.578$ T. The insets are zoomed-in views from the dashed black squares. **b** Left panels: time snapshots of voltage for applied currents of 100, 180, 220, 260, and 550 nA (from bottom to top). Right panels: the corresponding histograms of voltage distribution with a sampling time of 600 s. **c** The switching probabilities of the high and low voltage states as a function of the applied current $I$.

(e.g., COMS-based: $2 \times 10^{-4}$ W, conventional metastable MTJ-based: $1 \times 10^{-5}$ W). Specifically, with a tunneling current of 200 nA and a measured voltage of 0.45 V, the power consumption of our device is approximately $9 \times 10^{-8}$ W (Fig. 3). This finding suggests that 2D vdW magnet-based tunneling junction devices hold the potential for developing energy-efficient devices for advanced computing technologies.

**Unidirectional spin-state transition and stochastic switching in four-layer CrI₃**

We conduct similar measurements on other few-layer CrI₃ samples of varying thicknesses, including 2, 4, and 5L, as summarized in Supplementary Table 1, confirming that the remarkable characteristics observed are not exclusive to a particular thickness or device. Supplementary Fig. 12 depicts voltage as a function of applied bias current for a 4L CrI₃ tunnel junction device (~2.4 nm) (see Supplementary Figs. 13–15), measured at a magnetic field of 1.72 T and temperature $T = 1.5$ K. Similar to the 2L CrI₃ behavior (Fig. 2a), both bias current-controlled unidirectional spin-state transition at a lower tunneling current region and stochastic switching with relatively high tunneling currents are evident in the V-I characteristic. For example, multiple hysteresis loops in the V-I profile are observable at distinct bias current regions, attributed to tunneling current-induced various spin-state transitions among different magnetic domains (Supplementary Figs. 12a, 16a). The associated magnetic phases with different spin configurations in the magnetic domains are further validated through tunneling magnetoresistance (TMR) measurements at different bias currents (Supplementary Fig. 12b). Notably, in contrast to the 2L CrI₃, the 4L CrI₃ tunnel junction device displays stochastic switching with the involvement of three spin states. We note that this behavior is unattainable in metastable MTJs composed of conventional magnetic and insulating layers. Furthermore, both the unidirectional spin-state transition and stochastic switching appear in a much lower current range in the 4L CrI₃ tunnel junction device (Supplementary Figs. 12a, 16b) at $T = 1.5$ K. Supplementary Fig. 16c provides time snapshots of voltage measurements (left panels) at varied bias currents, along with the corresponding voltage distribution histograms (right panels). This data suggests that the three magnetic states in the 4L CrI₃ can also be effectively modulated by the tunneling current. This result reveals that the number of layers in 2D vdW magnets can serve as a new degree of freedom for the generation of novel magnetic states, an option not viable with traditional magnets. We further note that the unidirectional spin-state transition and stochastic switching have also been demonstrated in a 5L CrI₃ tunnel junction device (Supplementary Fig. 17).

**Understanding the tunneling current-induced unidirectional magnetization reversal in 2L CrI₃**

To understand why the direction of switching between SAP and SP states correlates with the polarity of the bias, we construct a minimal model of the experimental system that consists of two insulating magnetic layers sandwiched between two metallic non-magnetic layers, which resemble the 2L CrI₃ and the top- and bottom-most graphene layers in the graphite electrodes, respectively. To capture the main features of the CrI₃ tunnel barrier, we assume each insulating layer has two orbital and two spin degrees of freedom, schematically shown in Fig. 4a. The splittings between the orbital and spin states are chosen so that the Fermi energy is in the middle of the gap between same-spin but different-orbital states, which is the case for CrI₃[45]. That the majority spin states are closer to the Fermi energy than the minority ones is the main reason for the giant TMR of multilayer CrI₃[17]. The orbital and spin splittings are fixed based on first-principles calculations[46] and to reproduce the previously reported ~100% TMR in the linear response regime[16,47]. The four-layer

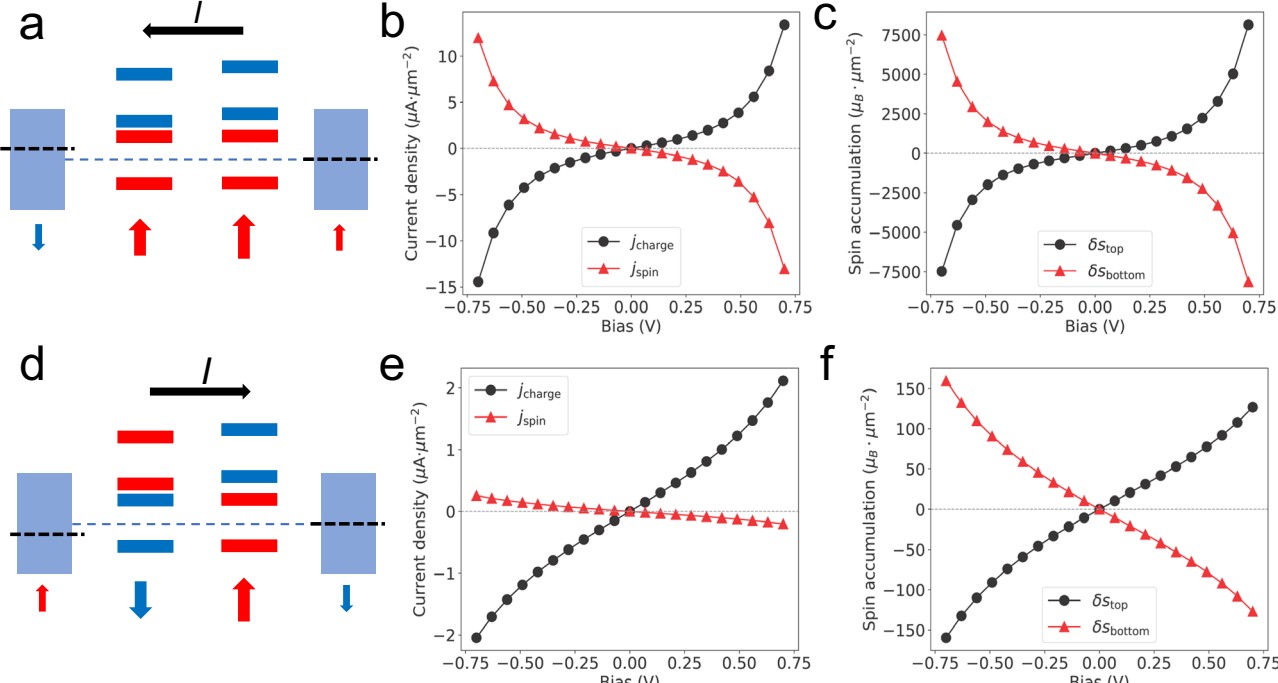

**Fig. 4 | Unidirectional tunneling-driven switching based on a minimal model.** **a** Schematic illustration of the tunneling-driven staggered spin accumulation in the SP state with a negative bias voltage (or positive δμ), with the left (right) side of the figure corresponding to the top (bottom) side of the actual device. The rectangular blocks represent the bands of the metal layers, with the black dashed lines standing for the chemical potential. The thick red (blue) horizontal lines represent the majority- (minority-) spin levels in the insulating layers. The large red vertical arrows stand for the direction of the magnetizations in the two CrI₃ layers, with the top layer having a smaller magnetization. The small red (blue) vertical arrows mean the accumulation of up (down) spins in the metal layers. **b** Tunneling charge- (black

dots) and spin current (red triangles) versus bias voltage in the SP state calculated using the NEGF approach. The spin current has the units of μA·(ℏ/2e). **c** Tunneling-induced spin accumulation in the top and bottom metal layers versus bias voltage, obtained by scaling the NEGF results by $\tau_s/\tau_s^{model}$ (Methods and Supplementary Information). **d** Schematic illustration of the tunneling-driven staggered spin accumulation in the SAP state with a positive bias voltage (negative δμ). **e**, **f** Tunneling charge- and spin current (**e**) and spin accumulation (**f**) versus bias voltage in the SAP state, with the top insulator layer having a 0.2 eV smaller spin splitting than that of the bottom insulating layer (2J = 2.8 eV).

model is then coupled with external leads with different chemical potentials to calculate the tunneling-related physical quantities using the Keldysh NEGF method (see Methods).

We first consider tunneling in the ferromagnetic configuration as the switching from SP to SAP states driven by the tunneling current is surprising within the conventional picture of STT, which usually favors the SP state. Figure 4b shows that the tunneling current is strongly spin-polarized due to the minority spin electrons seeing a much higher tunnel barrier than the majority spin ones. As a result, the majority of spin in the upstream metal (graphene) layer will decrease while that in the downstream layer will increase, as shown in Fig. 4c. The increase will eventually be balanced by spin relaxation processes that allow the system to reach a steady state. Assuming a spin-relaxation time of $10^2$ ps in graphene[48], a spin current density of 1 μA/μm² as in the present system can lead to a spin or magnetic moment accumulation of $\sim 6 \times 10^2$ μ_B/μm², or $\sim 2.5 \times 10^{-4}$ μ_B per CrI₃ unit cell. If the exchange coupling between the graphene electron spins and those in CrI₃ is on the order of $10^{-2}$ eV[47], it alters spin splittings by ~1 μeV per Cr moment. It has been estimated that the interlayer exchange coupling between the Cr moments in 2L CrI₃ is about 230 μeV[49]. However, the applied magnetic field of ~0.5–0.6 T almost balances out the energy difference between the SAP and SP configurations. As a result, a relatively small, staggered, exchange field due to the spin accumulation when the 2L CrI₃ is in the SP state can drive to the SAP state.

The above analysis seems to suggest that an increasing tunneling current can induce SP to SAP transition regardless of the bias polarity. We next show that this will not be the case if the two SAP states, ↑↓ and ↓↑, are not degenerate under a finite magnetic field. Asymmetry

between the local environments of the two CrI₃ layers in the tunnel junction is inevitable[18], and there is always a remnant net magnetization in the SAP state. In other words, the two CrI₃ layers do not have the same local moment density. A finite magnetic field, therefore, lifts the degeneracy between the ↑↓ and ↓↑ states. As a result, switching can take place only when the staggered spin accumulation due to the tunneling current is compatible with the low-energy SAP state under the magnetic field, as illustrated in Fig. 4a. The fact that the tunneling induced hysteretic SP to SAP transition only happens for negative bias suggests that the bottom CrI₃ layer has a larger magnetization than the top one, at least for the domain that is switched by the current.

The asymmetry between the two CrI₃ layers also explains the tunneling-driven SAP to SP transition, which happens only under a positive bias. If the two CrI₃ layers are identical except for having opposite magnetization directions in the SAP state, one can show that the tunneling spin current vanishes identically (Supplementary Note 2). However, any asymmetry between the two layers will, in general, render the tunneling spin current nonzero. Since we found above that the top layer has a smaller net magnetization which is expected to scale with the spin splitting seen by the tunneling electrons, we plot in Fig. 4e, f the NEGF results obtained by assuming the top CrI₃ layer to have a 0.2 eV smaller exchange splitting than that of the bottom layer, which is 2.8 eV. Other cases, including asymmetric orbital splitting, vertical dipolar field, and asymmetric hopping, are discussed in Supplementary Note 4. Figure 4e shows that such a small asymmetry leads to a tunneling spin current, about 10% in magnitude of the charge current. Since the tunneling current is kept fixed in the experiments despite the different resistance in the two states, the spin accumulation in the SAP state is

expected to be comparable to that in the SP state. More importantly, since experimentally, we always sweep the magnetic field from a saturating value in the same direction to the ones at which bias is swept, in the SAP state, the layer with a larger net magnetization is always aligned with the field. Therefore, the nonequilibrium staggered spin accumulation drives the SAP to SP transition only if the spin accumulation at the side of the smaller magnetization layer is opposite to its present direction, i.e., under positive bias, as illustrated in Fig. 4d.

The above scenario is consistent with the magnetic field dependence of the hysteretic tunneling-induced switching shown in Fig. 2 and Supplementary Fig. 6. On the larger magnetic field side of the phase boundary, where the SP state becomes favorable over the SAP state, the bias needed to induce the SP to SAP transition becomes more negative. This is because the relatively small size of the spin accumulation needs to overcome at least the energy difference between the SAP and SP states for the domains to be switched. On the other hand, there is no SAP and SP switching at positive bias, since, supposedly, the field is strong enough to eliminate AFM domains. In contrast, at the SAP side of the phase boundary, the critical bias for switching from SAP and SP becomes more positive as the field decreases so that the SAP state is more stable, while there is no SP to SAP switching at a negative bias. Finally, both SAP → SP and SP → SAP switching appears at intermediate field strengths, where both domains should exist. We note that although the deterministic magnetization switching was reported on a metallic 3D CoPt thin film[50], the underlying mechanism is SOT, which is fundamentally different from this work.

Our picture assumes that the magnetizations of the two CrI$_3$ layers are strictly collinear, which applies to the situation with a relatively low or moderate bias and low temperatures. At a higher bias beyond the linear response regime, heating induced by the tunneling current is significant. The large population of thermally activated magnons of the CrI$_3$ layers will make it more appropriate to model the CrI$_3$ magnetizations as noncollinear dynamical macrospins that are coupled with tunneling-induced spin torque in a self-consistent manner, which is more compatible with the conventional STT-induced transitions in ferromagnetic-metal-based MTJs and can be studied using similar methods[51,52]. This is further evidenced by observations that the stochastic switching at a higher bias does not have as strong asymmetry between opposite bias polarities as the low-bias hysteretic switching, and that the statistics of the transient states follow the bias size in a similar way as the stochastic magnetic tunnel junctions[41–44]. Finally, we note that additional kinetic effects induced by the tunneling currents may exist near magnetic domain walls of CrI$_3$, where there can be significant derivation of the Cr spin directions from that deep inside the domains. Such potential contributions, although interesting, rely more on the details of the domain walls and are not as deterministic as the mechanism explained above. Finally, we note that additional kinetic effects induced by the tunneling currents may exist near magnetic domain walls of CrI$_3$, where there can be significant deviation of the Cr spin directions from that deep inside the domains, or inside CrI$_3$ domains through the SOT[53]. Such potential contributions, although interesting, rely more on material details and are not as deterministic as the mechanism explained above.

Lastly, we discuss the potential influence of electric fields on the unidirectional spin-state transitions observed in our CrI$_3$ devices. Contrary to previous studies[18,19,54], our devices generate electric fields through the two graphene electrodes instead of top and/or bottom gates. While the magnitude of these electric fields is comparable to those produced by gate voltages in the earlier studies, the absence of top and/or bottom gates in our devices leads to a significantly reduced electrostatic doping effect. Notably, Jiang et al. demonstrated the layer spin-state transition in 2 L CrI$_3$ is primarily governed by electrostatic doping rather than an electric field[18]. This demonstration accounts for the absence of tunneling current-induced layer spin-state transitions in our CrI$_3$ devices. Furthermore, the spin-state transitions induced by

tunneling current in our devices are characterized by sudden, step-like changes in resistance/voltage, which stands in contrast to the more gradual magnetization switching typically associated with electrostatic doping[18,19,54]. Therefore, in agreement with prior research[18], we conclude that the role of electric fields in manipulating the spin states in our CrI$_3$ samples is minimal.

In summary, we report the pioneering observation of tunneling current-controlled spin-state transition among magnetic domains in atomically thin CrI$_3$ layers under magnetic fields near the layer-magnetic phase transition. Intriguingly, the associated transition between the SAP and SP states is deterministic, contrasting with STT, and exhibits a pronounced dependence on the tunneling current polarity. To account for the empirical findings, we develop a theoretical model predicated on tunneling current-induced spin accumulation, employing the Keldysh NEGF method. This model provides new insights into this phenomenon, suggesting broader applicability to other 2D magnetic materials. Furthermore, we reveal that at increased bias currents, the few-layer CrI$_3$ exhibits multi-state stochastic switching, and the switching probability is well controlled through the tunneling current, with power consumption orders of magnitude lower than traditional stochastic MTJs. Although currently limited to cryogenic operations and not an immediate alternative to room-temperature stochastic MTJs, our devices present a compelling alternative by addressing several fundamental challenges associated with conventional technologies, igniting future technological and scientific breakthroughs in this new direction, e.g., materials selection, fabrication refinements, and specialized applications. Our work, therefore, lays the foundation for an innovative approach to creating and manipulating magnetic states using 2D vdW magnets, which hold immense potential for advancing energy-efficient computing technologies, including nanoscale logic gates and probabilistic and neuromorphic computing systems.

## Methods

### CrI$_3$ single-crystal growth and device fabrication
CrI$_3$ single crystals were grown by a chemical vapor transport technique[55,56] using stoichiometric mixtures of Cr and I in a sealed evacuated quartz tube. The phase of the obtained crystals was checked by X-ray diffraction. Our few-layer CrI$_3$ tunnel junction devices were fabricated by the layer-by-layer dry-transfer method as detailed in refs. 30–32. Atomically thin flakes of hBN, graphite, and CrI$_3$ were mechanically exfoliated from the bulk crystals onto SiO$_2$(280 nm)/Si substrates. The thicknesses of the flakes are determined by their optical contacts as well as an atomic force microscope. The stack of hBN/graphite/CrI$_3$/graphite/hBN was picked up one by one using a polydimethylsiloxane stamp with a polyvinyl alcohol (PVA) layer on the top. The entire stack was then released onto a SiO$_2$/Si substrate with prefabricated Pt/Ti (30 nm/5 nm) electrodes, which were prepared by standard photolithography. The PVA layer on the device surface was dissolved in deionized water before the measurements. To avoid any degradation of the thin CrI$_3$ layers, the exfoliation and the transfer processes were performed in an argon-filled glove box with H$_2$O and O$_2$ concentrations of <0.1 ppm.

### Electrical and magnetotransport measurements
The electrical and magnetotransport measurements were performed inside a closed cycle $^4$He cryostat (Oxford TeslatronPT) with a base temperature of 1.5 K. The DC electrical transport measurements were carried out using a Keithley 2400 sourcemeter. Either a bias voltage or a bias current was used in the I-V measurements. The integration time used for these measurements is 16 ms. The measurements of time snapshots for voltage in Fig. 2d were carried out by a LeCroy Wavesurfer 452 oscilloscope with a 500 MHz bandwidth. In all the measurements, the ramping rate of the applied out-of-plane magnetic field is 1 mT/s. For all the I-V measurements under a background magnetic field, the field is achieved by first ramping to a saturated field (1.2 T for

2L CrI$_3$ and 2.5 T for 4L CrI$_3$), then decreasing to the target values. In our measurements, the I-V curves were ramped from zero to the most negative voltages, then to the most positive voltages, and finally back to zero. We note that all the measurements were taken from a single 2 L CrI$_3$ device and a single 4L CrI$_3$ device, respectively.

## Theoretical calculations of the tunneling spin current and nonequilibrium spin accumulation

We calculate the tunneling-induced effects by using the Keldysh NEGF technique applied to a minimal model motivated by a realistic CrI$_3$ bilayer sandwiched between graphite electrodes. The Hamiltonian is

$$H = H_L + H_D + H_T \tag{1}$$

where the three terms stand for Hamiltonians of the leads ($L$), the tunneling device ($D$), and the coupling between the two ($T$). The device consists of four layers as depicted in Fig. 4 of the main text. Two layers in the middle are insulating and magnetic, representing the 2L CrI$_3$. The two outermost layers are metallic, standing for graphene layers in the graphite electrodes that are in direct contact with the CrI$_3$. The device Hamiltonian is

$$
\begin{aligned}
H_D &= H_{M1} + H_{M2} + H_{I1} + H_{I2} + H_{MI1} + H_{MI2} + H_{I12} \\
&= \sum_{\mathbf{k}} \Big[ \sum_{l\sigma\tau} \epsilon_{\mathbf{k}}^M c_{l\sigma\mathbf{k}}^\dagger c_{l\sigma\mathbf{k}} + \sum_{l\sigma\tau} (\epsilon_{\mathbf{k}}^I - \Delta\tau - J_l\sigma - \mu_I) d_{l\sigma\mathbf{k}}^\dagger d_{l\sigma\mathbf{k}} \\
&\quad + \sum_{l\sigma\tau} t(c_{l\sigma\mathbf{k}}^\dagger d_{l\sigma\mathbf{k}} + \text{h.c.}) + \sum_{\sigma\tau} t(d_{1\sigma\mathbf{k}}^\dagger d_{2\sigma\mathbf{k}} + \text{h.c.}) \Big]
\end{aligned}
\tag{2}
$$

where M and I stand for metal and insulator layers, respectively, $\mathbf{k}$ is the 2D crystal momentum, $\sigma$ labels spin, $\tau$ labels orbital, $\Delta$ represents the orbital splitting, $J_{1,2}$ stand for the spin splitting in the two insulator layers, and $t$ is the hopping between neighboring layers. $H_D$ can also be written into a matrix form

$$H_D = \sum_{\mathbf{k}\sigma\tau} C_{\mathbf{k}\tau\sigma}^\dagger h_{\tau\sigma}(\mathbf{k}) C_{\mathbf{k}\tau\sigma} \tag{3}$$

where $C_{\mathbf{k}\tau\sigma} = (c_{1\tau\sigma\mathbf{k}}, d_{1\tau\sigma\mathbf{k}}, d_{2\tau\sigma\mathbf{k}}, c_{2\tau\sigma\mathbf{k}})^T$, and

$$
h_{\tau\sigma}(\mathbf{k}) = \begin{pmatrix}
\epsilon_{\mathbf{k}}^M & t & 0 & 0 \\
t & \epsilon_{\mathbf{k}}^I - \Delta\tau - J_1\sigma - \mu_I & t & 0 \\
0 & t & \epsilon_{\mathbf{k}}^I - \Delta\tau - J_2\sigma - \mu_I & t \\
0 & 0 & t & \epsilon_{\mathbf{k}}^M
\end{pmatrix}
\tag{4}
$$

Below we briefly summarize the standard procedure for applying the Keldysh formalism to tunneling problems. Hamiltonians of the form of Eq. 1 can in general be written as a matrix

$$H = \begin{pmatrix} H_L & T \\ T^\dagger & H_D \end{pmatrix} \tag{5}$$

The Keldysh contour-ordered 1-body Green's function satisfies

$$\begin{pmatrix} i\hbar\partial_t - H_L & -T \\ -T^\dagger & i\hbar\partial_t - H_D \end{pmatrix} \begin{pmatrix} G_L & G_{LD} \\ G_{DL} & G_D \end{pmatrix} = 1_c \tag{6}$$

where $1_c$ is an identity matrix in the space spanned by physical quantum numbers and time on the Keldysh contour. Inverting the block matrix gives

$$G_D = \Big[ i\hbar\partial_t - H_D - T^\dagger(i\hbar\partial_t - H_L)^{-1} T \Big]^{-1} \tag{7}$$

If we regard the leads to be a large non-interacting system with properties that are not affected by its coupling with the device, we have $(i\hbar\partial_t - H_L)^{-1} = g_L$, $g_L$ being the Keldysh Green's function of the leads by themselves. Then the effect of the leads is equivalent to a self-energy for $G_D$: $\Sigma_D = T^\dagger g_L T$. It then follows from the Dyson equation and the Langreth rules that the retarded, advanced, lesser, and greater Green's functions are

$$
\begin{aligned}
G_D^{r,a} &= \Big[ (g_D^{r,a})^{-1} - T^\dagger g_L^{r,a} T \Big]^{-1} \\
G_D^{<,>} &= G_D^r \Sigma_D^{<,>} G_D^a = G_D^r T^\dagger g_L^{<,>} T G_D^a
\end{aligned}
\tag{8}
$$

For fermion systems in equilibrium Green's functions in the eigenstate (labeled by $m$) and frequency representation are

$$
\begin{aligned}
g_m^{r,a}(\omega) &= (\hbar\omega - \epsilon_m^L \pm i\eta)^{-1}, \\
g_m^<(\omega) &= 2\pi i f(\epsilon_m^L) \delta(\hbar\omega - \epsilon_m^L), \\
g_m^>(\omega) &= 2\pi i [f(\epsilon_m^L) - 1] \delta(\hbar\omega - \epsilon_m^L)
\end{aligned}
\tag{9}
$$

The self-energies are therefore

$$
\begin{aligned}
(\Sigma_D^{r,a})_{jk}(\omega) &= \sum_m \frac{T_{jm}^\dagger T_{mk}}{\hbar\omega - \epsilon_m^L \pm i\eta} \\
(\Sigma_D^<)_{jk}(\omega) &= 2\pi i \sum_m T_{jm}^\dagger T_{mk} f(\epsilon_m^L) \delta(\hbar\omega - \epsilon_m^L) \\
(\Sigma_D^>)_{jk}(\omega) &= 2\pi i \sum_m T_{jm}^\dagger T_{mk} [f(\epsilon_m^L) - 1] \delta(\hbar\omega - \epsilon_m^L)
\end{aligned}
\tag{10}
$$

where $j,k$ label basis functions of $H_D$. As a first approximation, we assume that $\Sigma_D$ has only diagonal elements and is purely imaginary, corresponding to leads with trivial electronic structure. As a result,

$$
\begin{aligned}
\Sigma_D^{r,a}(\omega) &= \mp i\pi t^2 N_L \equiv \mp i\Gamma \\
\Sigma_D^<(\omega) &= 2\pi i t^2 N_L f(\hbar\omega) = 2i\Gamma f(\hbar\omega) \\
\Sigma_D^>(\omega) &= 2\pi i t^2 N_L [f(\hbar\omega) - 1] = 2i\Gamma [f(\hbar\omega) - 1]
\end{aligned}
\tag{11}
$$

where $t$ is the constant coupling parameter between the device and the leads, $N_L$ is the density of states of the leads at the Fermi energy, assumed to be a constant. Combining the self-energies with $H_D$ allows one to get the device's Green's functions according to Eq. 8. To calculate the expectation value of a 1-particle physical observable defined using single-particle states in the system, e.g.,

$$O = \sum_{jk} O_{jk} c_j^\dagger c_k \tag{12}$$

in nonequilibrium, we use

$$
\begin{aligned}
\langle O \rangle(t) &= \sum_{jk} O_{jk} \langle c_j^\dagger(t) c_k(t) \rangle = -i\hbar \sum_{jk} O_{jk} (G_D^<)_{kj}(t,t) \\
&= \hbar \int_{-\infty}^\infty \frac{d\omega}{2\pi} \text{ImTr}[O G_D^<(\omega)]
\end{aligned}
\tag{13}
$$

which is constant in the steady state.

For the device Hamiltonian Eq. 2, the self-energies, according to Eq. 11 are

$$
\begin{aligned}
(\Sigma_D^{r,a})_{\tau\sigma\mathbf{k}}(\omega) &= \mp i \begin{pmatrix} \Gamma & & & \\ & \eta & & \\ & & \eta & \\ & & & \Gamma \end{pmatrix} \\
(\Sigma_D^<)_{\tau\sigma\mathbf{k}}(\omega) &= 2i\Gamma \begin{pmatrix} f_1(\hbar\omega) & & & \\ & 0 & & \\ & & 0 & \\ & & & f_2(\hbar\omega) \end{pmatrix}
\end{aligned}
\tag{14}
$$

where $\eta = 0^+$ is included to ensure that the bare Green's functions of isolated insulator layers have the correct analytical properties, $f_1$ and $f_2$ are the Fermi-Dirac distribution functions for the two leads in contact with the top and bottom metal layers, respectively. The only difference between $f_1$ and $f_2$ is that they have different chemical potentials $\mu_1$ and $\mu_2$. We assume the bottom ($l = 2$) layer is grounded and has its chemical potential $\mu_2 = 0$. A finite $\mu_1 \equiv \delta\mu$ then corresponds to a finite bias potential.

From the above results, we obtain the retarded and advanced Green's functions for the four-layer system

$$(G_D^{r,a})_{\tau\sigma\mathbf{k}}(\omega) = \begin{pmatrix} \hbar\omega - \epsilon_{\mathbf{k}}^M \pm i\Gamma & -t & 0 & 0 \\ -t & \hbar\omega - \epsilon_{\mathbf{k}}^l + E_{I1} \pm i\eta & -t & 0 \\ 0 & -t & \hbar\omega - \epsilon_{\mathbf{k}}^l + E_{I2} \pm i\eta & -t \\ 0 & 0 & -t & \hbar\omega - \epsilon_{\mathbf{k}}^M \pm i\Gamma \end{pmatrix}^{-1}$$

(15)

where

$$E_{I1,2} \equiv \Delta\tau + J_{1,2}\sigma + \mu_I \qquad (16)$$

The above $G_D^{r,a}$ together with Eq. 14 give $G_D^<$ through Eq. 8.

The main observables that we calculate are spin/charge currents and nonequilibrium spin accumulation. For a tight-binding model, the electric current density operator is ($e$ is the absolute value of the electron charge)

$$\mathbf{j} = -e\mathbf{v} = -\frac{e}{i\hbar}[\mathbf{r}, H] = \frac{e}{i\hbar}\sum_{ij}(\mathbf{r}_i - \mathbf{r}_j)t_{ij}c_i^\dagger c_j \qquad (17)$$

For the present model, since current only flows from one lead to the other, we can regard

$$\hbar\int_{-\infty}^{\infty}\frac{d\omega}{2\pi}\mathrm{ImTr}[\mathbf{j}\cdot\hat{n}G_D^<(\omega)] = Id \qquad (18)$$

where $d$ is a dimensionless integer that stands for the thickness of the junction and $\hat{n}$ is a unit vector normal to the junction. Namely, the spin- and orbital-resolved electric current operator $I$ is

$$I_{\tau\sigma} = -\frac{e}{i\hbar d}\begin{pmatrix} 0 & -t & 0 & 0 \\ t & 0 & -t & 0 \\ 0 & t & 0 & -t \\ 0 & 0 & t & 0 \end{pmatrix} \qquad (19)$$

where $d = 3$. Alternatively, the above formula can be understood as the average tunneling current between each two neighboring layers and is equivalent to the Landauer formula for the present model in which the tunneling is fully coherent. The spin current is accordingly

$$I_s = -\frac{\hbar}{2e}\sum_\tau(I_{\tau\uparrow} - I_{\tau\downarrow}) \qquad (20)$$

Therefore the charge and spin currents have opposite signs assuming electron-type carriers, consistent with Fig. 4b,e. The dimension of the spin current is also chosen as $\mu$A with the understanding that it is to be multiplied by $\hbar/2e$. The nonequilibrium spin accumulation is calculated similarly by using Eq. 13 minus the local spin density in equilibrium. In Fig. 4, we have rescaled the spin accumulation by $\tau_s/\tau_s^{\text{model}}$, where $\tau_s$ is the experimental spin relaxation time, and $\tau_s^{\text{model}} = \frac{\hbar}{2\Gamma}$ is the spin relaxation time of our toy model calculation (see Supplementary Information).

To perform the integration over $\mathbf{k}$, since the integrand in general depends on $\mathbf{k}$ through both $\epsilon_{\mathbf{k}}^M$ and $\epsilon_{\mathbf{k}}^l$, one cannot replace the 2D momentum integral by a 1D integral over energy multiplied by a

density of states function. In fact, such a 2D momentum dependence of the spin-resolved tunneling amplitude determined by the electronic structures of both the metal and the insulator layers is the key to the giant tunneling magnetoresistance in Fe/MgO/Fe tunnel junctions, known as the symmetry filtering effect. In our toy model, however, it is not realistic to fully account for such details in the real graphene/CrI$_3$ junctions. We simply choose $\epsilon_{\mathbf{k}}^l \equiv w\epsilon_{\mathbf{k}}^M$, where $w \ll 1$ is a scaling factor accounting for the fact that the insulator's bandwidth is much smaller than that of the metal layers. In this way, we can do the following

$$\int\frac{d^2\mathbf{k}}{(2\pi)^2}F(\epsilon_{\mathbf{k}}^M, \epsilon_{\mathbf{k}}^l) = \int\frac{d^2\mathbf{k}}{(2\pi)^2}F(\epsilon_{\mathbf{k}}^M, w\epsilon_{\mathbf{k}}^M) = \int d\epsilon N_M(\epsilon)F(\epsilon, w\epsilon) \qquad (21)$$

Moreover, if we stay away from the resonant tunneling regime to be consistent with experiment, we may ignore $\epsilon_{\mathbf{k}}^l$, which is equivalent to setting $w = 0$ in Eq. 21. For a given physical quantity $O$ its expectation value will be

$$\langle O \rangle = \frac{1}{2\pi}\int_{-\infty}^{\infty}d\epsilon'\int_{-\infty}^{\infty}d\epsilon N_M(\epsilon)\mathrm{ImTr}[OG_D^<(\epsilon', \epsilon, \delta\mu)] \qquad (22)$$

where we have defined $\epsilon' \equiv \hbar\omega$.

For simplicity assume that the density of states $N_M(\epsilon)$ is a constant $N_F$ in the range of $[-W/2, W/2]$, which limits the $\epsilon$ integral in the same range. More specifically, for current $I$, at low temperature when $f_{1,2}$ can be approximated by step functions, we use

$$\langle I \rangle = \frac{N_F}{2\pi}\int_{\min(0,\delta\mu)}^{\max(0,\delta\mu)}d\epsilon'\int_{-W/2}^{W/2}d\epsilon\,\mathrm{ImTr}[IG_D^<(\epsilon', \epsilon, \delta\mu)], \qquad (23)$$

At a given $\epsilon'$, the integrands as functions of $\epsilon$ have poles only at $\epsilon'$, which provides the dominant contribution to the integral. Therefore we expect the result to be weakly dependent on $W$ as long as $W/2 > |\delta\mu|$. Once the integration over $\epsilon$ is calculated, the $\epsilon'$ integral will be regular except at the poles $\epsilon = E_{I1,2}$. This will not be a concern since we do not consider the resonant tunneling regime. Namely, $-E_{I1,2}$ are not in the energy window $[\min(0,\delta\mu), \max(0,\delta\mu)]$.

The parameters in our model are listed in Supplementary Table 2. The parameter values are set based on comparison with experimental or theoretical facts related to CrI$_3$ and graphene. Since the quasiparticle gap is over 2 eV[46] and it is between the same-spin but different-orbital states, we choose $J = 1.4$ eV and $\Delta = 0.8J = 1.12$ eV. In the linear response regime we find that $\Delta/J = 0.8$ leads to a tunneling magnetoresistance $\sim 100\%$ as reported in previous experiments. The interlayer hopping $t$ is chosen to be 0.03 eV according to DFT results[57] and we set it to be uniform across the junction. For graphene on hBN, if taking the carrier density to be $10^{12}$ cm$^{-2}$, the Fermi momentum is $k_F \sim 10^{-2}$ Å$^{-1}$ and the Fermi energy $E_F \sim 0.1$ eV. Thus $N_F \sim 0.01$ eV$^{-1}$ per unit cell, or $\sim 2 \times 10^5$ eV$^{-1}\mu$m$^{-2}$. We take the sample area to be 1 $\mu$m$^2$. Moreover, in our model $N_F$ is the density of states per spin per orbital. So we set $N_F = 5 \times 10^4$ eV$^{-1}$ for a 1 $\mu$m$^2$ sample. To estimate $\Gamma$, we note that the interlayer hopping in graphite is on the order of $0.1 - 0.3$ eV[58]. Using the same density of states above leads to $\Gamma = \pi t^2 N_F \approx 0.003$ eV. For $W$, as mentioned above it should not have a strong influence on the results as long as it is much greater than $|\delta\mu|$. We have used a $W = 2$ eV and found that a larger $W$ does not lead to significant change of the results. Finally the spin relaxation time $\tau_s$ is based on experimental values reported in ref. 48

In addition, to capture the different exchange splittings of the two CrI$_3$ layers as mentioned in the main text, we set the exchange coupling of the bottom (top) layer $|J_2| = J$ ($|J_1| = J - 0.1$ eV), and the chemical potential $\mu_{I1,2} = -|J_{1,2}|$ so that the Fermi energy is still at the center of the gap of each insulator layer (Supplementary Fig. 18). The value of 0.1 eV was chosen arbitrarily to show the qualitative consequences of

the symmetry breaking. The other possible ways that make the two layers asymmetric are discussed in Supplementary Information.

## Data availability
All relevant data are reported in the manuscript and in the associated Supplementary Information. The data that support the findings of this study are available from the corresponding authors on reasonable request.

## Code availability
The code is available upon reasonable request from the corresponding authors.

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

## Acknowledgements

This research was mainly supported by the US Department of Energy, Office of Basic Energy Sciences, Division of Materials Sciences and Engineering under award no. DE-SC0020074 (T.Y.C.) for sample and device fabrication and no. DE-SC0021281 (J.T.) for transport measurements. J.T. also acknowledges the financial support of the Wyoming NASA EPSCoR under award no. 80NSSC19M0061 and US National Science Foundation (NSF) grant 2228841 for data analysis, and the US NSF through the Penn State 2D Crystal Consortium Materials Innovation Platform (2DCC-MIP) under NSF cooperative Agreement no. DMR-2039351 for sample growth. Z.M. also acknowledges the support from NSF under Grant No. DMR 2211327. H.C. was partially supported by US NSF CAREER grant DMR-1945023. M.W. was supported by US NSF grant ECCS-1915849. K.W. and T.T. acknowledge support from the JSPS KAKENHI (Grant Numbers 20H00354 and 23H02052) and World Premier International Research Center Initiative (WPI), MEXT, Japan.

## Author contributions

J.T. conceived the project. Z.F. fabricated the devices. Z.F. and P.I.S. performed the measurements. J.A., Y.Z., and Z.M. grew the bulk CrI$_3$ crystals. K.W. and T.T. grew the bulk hBN crystals. A.H.M. and H.C. performed theoretical analysis. Z.F., W.W., Y.D, T.C., J.T., M.Z.W., A.H.M., H.C., and J.T. analyzed the data. All authors discussed the results and commented on the manuscript.

## Competing interests

The authors declare no competing interests.
