## [Peer Review File · Nature Communications]

Reviewers' Comments:

Reviewer #1:

Remarks to the Author:

This manuscript reports systematic investigations on electronic transport properties of few layer CrI₃ tunnel junctions. The authors find hysteresis behavior in IV curves when certain magnetic field is applied, and take this behavior as the evidences of spin states being manipulated by the tunneling current. With the support of theoretical calculations, this phenomenon is interpreted with a mechanism involving non-equilibrium spin accumulation in the graphene electrodes during the tunneling process. The authors further show the stochastic switching between spin states at certain range of tunneling current.

The investigations of 2D magnets are establishing new paradigm for spintronics, and how to effectively control spin states of 2D magnet through electric method is key task in this research field. The possibility of manipulating spin state with tunneling current and its underlying mechanism reported in this manuscript sounds very interesting. But I need the authors clarify following issues before I make the decision of recommendation of publication or not:

1. The manuscript emphasizes the unidirectional magnetic switching induced by tunneling current, meaning switching direction between SP and SAP determined by the polarity of the bias. I get confused by this claim, if we look at the right inset of Fig.2a, when the bias sweep from low current to high current (blue line), there is SAP to SP transition at around 3 μ A; and when the bias sweep from high current to low current (red line), there is SP to SAP transition at around 2 μ A. This means SP and SAP can be switched in both way at positive bias, not unidirectional switching as claimed in manuscript.

2. I also get confused by the description of main text and the figures, for example, it says "transition from SAP to SP configurations, ***, is invariably detected at positive bias current", but it does not show up at $B=0.586$ T as well as $B=0.566$ T (Fig.2c). The main text says "the transition from the SP to SAP configuration, ***, is manifested at negative bias current", but in the right inset of Fig.2c ($B=0.566$ T, red line), there is obviously SP to SAP transition at positive bias when current change from high to low value.

3. The magnetization of bilayer CrI₃ is uncompensated and result in large ferromagnetic-like hysteresis (Fig.1d), does the history of external magnetic field affect the behavior of SAP/SP switching? If yes, the authors should specify how the magnetic field is applied, for example, $B=0.550$ T is achieved from large positive (negative) magnetic field.

4. The authors proposed the switching mechanism involving non-equilibrium spin accumulation in the graphene electrodes. While I believe spin accumulation would indeed be there in graphene electrodes, it is not clear for me why the spin state prefer to change, take Fig.4a for example, the amplitude of spin accumulation in both graphene is the same and the spins/magnetic moment are colinear with CrI₃ layers, I do not see there is force/torque which initiates the spin state change from SP to SAP.

5. Within the proposed theoretical model, the authors argued that SAP to SP transition only happen at positive bias, because "the layer with a larger net magnetization is always pinned by the external magnetic field". This statement is not correct because the direction of larger net magnetization layer also depends on the history of applied magnetic field: if sweeping field from -1T to small positive magnetic field, the direction of larger net magnetization layer should be pointing to negative direction.

6. The authors emphasize the spins in this model are colinear, different from STT/SOT. However, what controlled in this work is one small domain in the sample. That means there are domain walls, whose volume can not be ignored as the domain itself is quite small (probably tens of nanometer in diameter), and the spins inside domain wall are not colinear.

7. The authors claimed that the switching current density and power is four orders smaller than conventional method of STT/SOT. But what controlled in this study eventually is one domain which is only small part of whole flake. The authors should find one way to estimate the area of this domain (such as from analysis of transport data) and get more accurate switching current density, or at least point this out.

8. The authors should discuss the electric field and doping effect on the magnetism of the sample, and exclude them as the origin of spin state change if possible. From extended Figure 3 and 4, the effect of electric field/doping on magnetism is large, they can modify the interlayer magnetic

interaction and net magnetization of bilayer samples, these two factors can also result in spin state change.

9. The authors should also discuss the effect of Joule heat as the current is not small, the increase of local temperature can result in the decrease of spin-flip field and the change of spin state.

10. One minor point, please identify which electrode is grounded in Fig.4a/d.

Reviewer #2:

Remarks to the Author:

This manuscript describes current-induced magnetization reversal (CIMR) in CrI₃ van der Waals heterostructure devices. CrI₃ is 2D magnet, which attracts much interest. The authors fabricated a device using double-layer CrI₃ with attaching graphene as the top and the bottom electrodes and demonstrated CIMR in the device. I show the following comments to the contents of the manuscript, which could contribute to evaluate the manuscript.

1. The concept of CIMR is well-established by many previous studies, and the authors' study does not show any significance and novelty in this perspective. In fact, novel physics of the CIMR in the CrI₃ devices is not included in this work. Hence, I found difficulty in finding any surprising and/or intriguing contents that warrants publication.

2. The weakness of the device is its high base resistance of the order of MegaOhm. Although the authors claimed energy efficient stochastic switching, this quite high base resistance strongly hinders applications. Thus, I do not agree with the authors' assertion that the device enables high density energy-efficient nanoscale logic gates and probabilistic/neuromorphic computing.

3. Whereas the authors claimed the stochastic switching is significant, it is merely due to uncontrolled magnetic states (magnetic domains, total magnetization, coercive fields etc...), I think.

4. Although the authors paid strong dedication to understand the underlying physics of the stochastic switching by combining with theory and experiments, missing and appearance of the symmetric switching at 1.5 K and 30 K was not sufficiently clear to me. Furthermore, if the authors' claim is current, the appearance of the switching can be in the positive bias in the other device because the net magnetization (magnetism) of two CrI₃ layers can be changed, and the probability of the appearance can be close to 50%. I also point out that the device characteristics are not fully controlled, and the authors implemented analyses only to a specific device, which hampers drawing and understanding of the whole picture of the observation if it is in this work.

In summary, I do not find sufficient significance and novelty that makes readers surprising in the manuscript at this stage. The authors tried to understand what takes place in their experiments. However, the device performance depends on the chance combination of the two CrI₃, which does not allow general understanding of profound and novel physics beyond the emblematic case of the authors' experiments.

Response to Referee #1:

Comment #1: This manuscript reports systematic investigations on electronic transport properties of few layer CrI₃ tunnel junctions. The authors find hysteresis behavior in IV curves when certain magnetic field is applied, and take this behavior as the evidences of spin states being manipulated by the tunneling current. With the support of theoretical calculations, this phenomenon is interpreted with a mechanism involving nonequilibrium spin accumulation in the graphene electrodes during the tunneling process. The authors further show the stochastic switching between spin states at certain range of tunneling current. The investigations of 2D magnets are establishing new paradigm for spintronics, and how to effectively control spin states of 2D magnet through electric method is key task in this research field. The possibility of manipulating spin state with tunneling current and its underlying mechanism reported in this manuscript sounds very interesting. But I need the authors clarify following issues before I make the decision of recommendation of publication or not:

Response: We thank the reviewer for the very positive perspective on our research topic, noting, "*The investigations of 2D magnets are establishing new paradigm for spintronics and how to effectively control spin states of 2D magnet through electric method is key task in this research field*" and rating our work as, "*...in this manuscript sounds very interesting.*" We have carefully addressed the reviewer's questions/comments below and look forward to his/her further consideration.

Comment #2: The manuscript emphasizes the unidirectional magnetic switching induced by tunneling current, meaning switching direction between SP and SAP determined by the polarity of the bias. I get confused by this claim, if we look at the right inset of Fig.2a, when the bias sweep from low current to high current (blue line), there is SAP to SP transition at around 3 μ A; and when the bias sweep from high current to low current (red line), there is SP to SAP transition at around 2 μ A. This means SP and SAP can be switched in both way at positive bias, not unidirectional switching as claimed in manuscript.

Response: We appreciate the reviewer highlighting this potential source of confusion. We concur with the reviewer's observation that sweeping the current back and forth allows the spin states to transition from SAP to SP and revert to SAP-exemplified within the positive current range-thus revealing the observed hysteresis loop. In the manuscript, we have defined the unidirectional spin-state transition as "*... (ii) as the amplitude of the tunneling currents increases, the transition from the SAP to SP configuration, corresponding to a high-to-low voltage state, is invariably detected at positive bias currents. In contrast, the transition from the SP to SAP configuration, characterized by a low-to-high voltage state, is manifested at negative bias currents. This observation represents the demonstration of tunneling current-induced unidirectional spin-phase transition in 2D vdW magnets.*" Specifically, as the current amplitude increases at positive values, the switching transitions from SAP to SP. Conversely, the spin state switching reverses

direction, from SAP to SP, within the region of negative current. It is important to note that we do not consider or reference the history of current sweeping. For instance, the switching direction is from SAP to SP at approximately $3 \mu\text{A}$ under positive bias current and the switching direction reverts from SP to SAP around $-0.08 \mu\text{A}$ under negative bias current, as depicted in Fig. 2a (and also in Fig. R1) in the revised manuscript. To preclude further confusion, we have refined the definition of the observed unidirectional spin-state transition in the revised manuscript and incorporated the inset of Fig. R1 into Fig. 2a to illustrate that such a tunneling current can induce spin state transition at much smaller currents.

Fig. R1 Voltage as a function of applied current of the 2L CrI_3 tunnel junction device at $B = 0.550 \text{ T}$. The inset shows the zoom-in of selected box. Reproduced from the data in Fig. 2a in the main text.

Comment #3: I also get confused by the description of main text and the figures, for example, it says "transition from SAP to SP configurations, ***, is invariably detected at positive bias current", but it does not show up at $B=0.586 \text{ T}$ as well as $B=0.566 \text{ T}$ (Fig.2c). The main text says "the transition from the SP to SAP configuration, ***, is manifested at negative bias current", but in the right inset of Fig.2c ($B=0.566 \text{ T}$, red line), there is obviously SP to SAP transition at positive bias when current change from high to low value.

Response: We appreciate the reviewer's comment. We surmise that the existing confusion is intertwined with the definition of unidirectional spin-state transition. As clarified in response to comment #2, our definition of unidirectional spin-state transition exclusively considers the polarity and amplitude of the applied bias current, excluding the current sweeping history. Once this definition is accepted, we believe that the reviewer can understand better and agree with our statement that "transition from SAP to SP configurations, ***, is invariably detected at positive bias current" and "the transition from the SP to SAP configuration, ***, is manifested at negative bias current" as stated in the manuscript.

We agree that the SAP to SP transition on the positive current range is missing in the data measured at $B=0.586$ T (Fig. 1c). This absence primarily arises as the current-induced spin state switching is profoundly influenced by the applied magnetic fields, evident in Extended Data Figs. 5-7. In this work, we have demonstrated that the background magnetic field enables the control of transitions between SAP and SP, allowing them to occur either within the positive or negative current ranges or within both, as depicted in Fig. 2 and Extended Data Figs. 5-7. Nonetheless, the observation remains valid: any witnessed transition from SAP to SA occurs invariably in the positive current range, while the transition from SP to SAP is solely observed within the negative current range.

Comment #4: The magnetization of bilayer CrI_3 is uncompensated and result in large ferromagnetic-like hysteresis (Fig. 1d), does the history of external magnetic field affect the behavior of SAP/SP switching? If yes, the authors should specify how the magnetic field is applied, for example, $B=0.550$ T is achieved from large positive (negative) magnetic field.

Response: We extend our thanks to the reviewer for highlighting this crucial aspect that was absent from our initial manuscript. We concur with the reviewer's notion that the history of the magnetic field does influence the initial spin state in the CrI_3 thin layers, which is evidenced by the hysteresis loop shown in Fig.1d. In our measurements, we initially sweep the magnetic field to a substantial saturation field (1.2 T for 2L CrI_3 and 2.5 T for the 4L CrI_3) to fully magnetize the CrI_3 layer before sweeping the magnetic field back to the desired target field. For instance, for measurements conducted at $B = 0.550$ T, the magnetic field was first swept from zero to 1.2 T, and subsequently, it was reduced to 0.550 T. To enhance clarity and completeness, we have incorporated sentences in the Methods section of the revised manuscript, detailing our approach to achieving the magnetic field.

Comment #5: The authors proposed the switching mechanism involving nonequilibrium spin accumulation in the graphene electrodes. While I believe spin accumulation would indeed be there in graphene electrodes, it is not clear for me why the spin state prefer to change, take Fig.4a for example, the amplitude of spin accumulation in both graphene is the same and the spins/magnetic moment are colinear with CrI_3 layers, I do not see there is force/torque which initiates the spin state change from SP to SAP.

Response: As explained in the main text, when the bilayer CrI_3 is driven to be close to the SP/SAP phase boundary by the external magnetic field so that the state that the system is currently in becomes metastable, a small perturbation favoring the target final state can make the transition occur if overcoming the potential energy barrier in-between the two states. In the case of Fig. 4a, the CrI_3 bilayer is in a metastable SP state, while the exchange coupling with the accumulated spin in the graphene electrodes favors the SAP state. When the latter plus thermal energy overcomes the energy barrier between the SP and SAP states, a transition to the SAP state will happen. Although the torque due to exchange fields that are strictly collinear

with the Cr spins vanishes, a slight deviation of the Cr spin directions from the averaged equilibrium orientation due to thermal fluctuations is sufficient to trigger the switching event. A similar process happens in the switching of a ferromagnet by an external magnetic field antiparallel with the magnetization as well.

We also kindly remind the referee that the switching we are considering is from SP to SAP, not from SP to its time reversal (flipping the moments of both CrI₃ layers together). The latter would not be helped by the staggered spin accumulation in the graphene electrodes, and is also strongly disfavored due to the applied magnetic field.

Comment #6: Within the proposed theoretical model, the authors argued that SAP to SP transition only happen at positive bias, because "the layer with a larger net magnetization is always pinned by the external magnetic field". This statement is not correct because the direction of larger net magnetization layer also depends on the history of applied magnetic field: if sweeping field from -1T to small positive magnetic field, the direction of larger net magnetization layer should be pointing to negative direction.

Response: We thank the reviewer for the thoughtful question and kindly refer them to our reply to Question #4. The SAP state at which we observe the tunneling-induced transition is obtained by sweeping the field down from a large saturating field to the desired value in the same direction, so the layer with larger magnetization should always be along the direction of the external field.

Comment #7: The authors emphasize the spins in this model are colinear, different from STT/SOT. However, what controlled in this work is one small domain in the sample. That means there are domain walls, whose volume can not be ignored as the domain itself is quite small (probably tens of nanometer in diameter), and the spins inside domain wall are not colinear.

Response: We thank the reviewer for raising this important point. In an alternative language, the stepwise switching events observed in the tunneling experiments correspond to discrete jumps of domain walls among different pinning centers as the domain walls sweep through the sample. The mechanism in our paper corresponds to a uniform, nonequilibrium, bias field in the whole area of the relevant domains, similar to the role of a uniform magnetic field in driving domain wall propagation when switching a ferromagnet. To which direction the domain wall will propagate is tied to which side of it being the favored phase subject to the bias field. We agree with the referee that at the domain wall the spins between the two CrI₃ layers may not be collinear anymore and the other kinetic effects, such as STT or SOT, of the tunneling currents can further help with the domain wall de-pinning and propagation, but such processes are not as deterministic as the one mentioned above. We have edited the relevant sentences in the main text to reflect the above discussion. We have also removed the sentence in the introduction contrasting the collinearity in the present system with the situation in conventional STT devices to avoid misunderstanding.

Comment #8: The authors claimed that the switching current density and power is four orders smaller than conventional method of STT/SOT. But what controlled in this study eventually is one domain which is only small part of whole flake. The authors should find one way to estimate the area of this domain (such as from analysis of transport data) and get more accurate switching current density, or at least point this out.

Response: We appreciate the reviewer's insightful comment. Indeed, our previously employed method to calculate the switching current density ($\sim 20 \text{ A/cm}^2$ for a tunneling current of $\sim 100 \text{ nA}$ and a junction area of $0.5 \mu\text{m}^2$)-by dividing the tunneling current by the junction area-doesn't reflect the actual physical picture as anticipated. According to the reviewer's suggestion, we have recalculated the switching current density by dividing the tunneling current by the estimated area of a single domain in CrI_3 by directly analyzing the transport data in Fig. 2a. To estimate the area of the magnetic domain and the current responsible for the spin state transition, we assume that the Ohm's law is valid, and the resistances of the magnetic domains with SP and SAP states connected in parallel in the circuit as shown in Fig. R2. Figure R2a-c depicts the simplified circuits of the 2L CrI_3 tunnel junction device displaying a layer SP state, a layer SAP state, and two magnetic domains with both SP and SAP states, respectively. As demonstrated in Fig. 2b, near the SAP to SP transition region, the layer magnetic state of the 2L CrI_3 is SAP when the magnetic field is below 0.2 T and shifts to SP when it surpasses 0.7 T. Figure 2b reveals that the current-induced transition from the SAP state (marked as ①) to the SP state (marked as ②) can be interpreted as the SAP state (with two magnetic domains in the layer) flipping to SP (with only one magnetic domain now). It's worth mentioning that there may be one or two additional small domains. However, their influence on the resistance change is minimal compared to the SAP to SP transition highlighted in the inset of Fig. 2a. Now, we can use the data in Fig. R2d to estimate the area of the magnetic domain with the SAP state. When the magnetic field is above 0.7 T, the layer magnetic state is SP, and we have $R_{SP} = \frac{\rho_{SP}L}{A}$, where R_{SP} is the resistance of the junction, A is the junction area ($\sim 0.5 \mu\text{m}^2$), L is the thickness of the sample, and ρ_{SP} is the corresponding resistivity. At 0.7 T (see Fig. R2d), we have $\rho_{SP}L = R_{SP}A = 0.5490 \text{ M}\Omega \times 0.5 \mu\text{m}^2 = 0.2745 \text{ M}\Omega \cdot \mu\text{m}^2$. Similarly, at 0.2 T (Fig. R2d), the layer magnetic state is SAP, and we have $R_{SAP} = \frac{\rho_{SAP}L}{A}$, where R_{SAP} and ρ_{SAP} are the resistance and resistivity of the tunnel junction, respectively. Then, $\rho_{SAP}L = R_{SAP}A_{SAP} = 0.603 \text{ M}\Omega \times 0.5 \mu\text{m}^2 = 0.3015 \text{ M}\Omega \cdot \mu\text{m}^2$. In the case of the intermediate state with one SP domain and one SAP domain (Fig. R2c) in the 2L CrI_3 , the total resistance R' of the intermediate state is $\frac{1}{R'} = \frac{1}{R'_{SP}} + \frac{1}{R'_{SAP}}$, where R'_{SP} and R'_{SAP} are the resistances of the SP and SAP domains, respectively. Then, we have $\frac{1}{R'} = \frac{1}{\frac{\rho_{SP}L}{A-A_{SAP}}} + \frac{1}{\frac{\rho_{SAP}L}{A_{SAP}}}$, where A_{SAP} is the area of the domain with an SAP state. From Fig. R2d, we know at state ①, $R' = 0.573 \text{ M}\Omega$, resulting in the area (A_{SAP}) of the magnetic domain with an SAP state of

$\sim 0.235 \mu\text{m}^2$. At state ② and $R' = 0.556 \text{ M}\Omega$, the estimated area of the magnetic domain is $\sim 0.072 \mu\text{m}^2$. Thus, the area of the magnetic domain that got switched from ① to ② is $\sim 0.163 \mu\text{m}^2$, which is comparable with the areas of the magnetic domains of CrI_3 thin layers captured by single-spin microscopy, as referenced in ref. 35. Next, we calculate the switching current, I_{SAP} , which is responsible for the SAP to SP transition at the bias current I of $3 \mu\text{A}$. From Fig. R2d, we have $I_{SAP} = \frac{IR'_{SP}}{R'_{SP} + R'_{SAP}} = 1.18 \mu\text{A}$. Thus, the switching current density ($\frac{I_{SAP}}{A_{SAP}}$) is $\sim 724 \text{ A/cm}^2$, which is still three orders of magnitude lower than the values reported in previous studies employing SOT. We note that the transition between SAP and SP states can be realized at a much lower current of $\sim 100 \text{ nA}$, and the switching current density could be even lower.

Fig. R2 The simplified circuits of the 2L CrI_3 tunnel device with (a) layer SP state, (b) layer SAP state, and (c) two domains with both SP and SAP states. (d) Tunneling resistance as a function of the magnetic field at two bias currents of $1 \mu\text{A}$ (top panel of Fig. 2a).

We have updated the switching current density in the revised manuscript and added the related discussion in Supplementary Note 8.

Comment #9: The authors should discuss the electric field and doping effect on the magnetism of the sample, and exclude them as the origin of spin state change if possible. From extended Figure 3 and 4, the effect of electric field/doping on magnetism is large, they can modify the interlayer magnetic interaction and net magnetization of bilayer samples, these two factors can also result in spin state change.

Response: We agree with the reviewer that the electric field/doping can strongly affect the magnetism of CrI_3 . Previous reports (Nat. Mater. 17, 406–410 (2018); Nat. Nanotechnol. 13, 549–553 (2018); Nat. Nanotechnol. 13, 544–548 (2018)) have demonstrated the control of magnetism in CrI_3 by electric field or doping under a background magnetic field. We note that all the works were realized via a top or back gate and no unidirectional magnetism state switching was observed. Furthermore, our CrI_3 tunnel junction device doesn't have either a top or back gate. Given the small bias voltage (300 to 400 mV) and the relatively large tunneling current (~ 50 to 400 nA) shown in the extended Figures 3 and 4, we believe that the effect

from the electrical field/doing should be minimal. Certainly, the reviewer pointed out our future research direction of studying how the electric field/doing affects the unidirectional spin state switching and stochastic switching in our devices. We have added the related discussion in the revised manuscript.

Comment #10: The authors should also discuss the effect of Joule heat as the current is not small, the increase of local temperature can result in the decrease of spin-flip field and the change of spin state.

Response: We agree with the reviewer's comment that the Joule heat plays a certain role in this work. In particular, one of the important sources of thermal noise needed for realizing stochastic switching is Joule heat induced by the tunneling current. However, we believe that the Joule heat plays little role in realizing the observed unidirectional spin state switching. We know that from Joule's law ($P = I^2R$), the Joule heat is determined by the magnitude of the current and independent of its polarity. However, the observed unidirectional spin state switching is not only dependent on the magnitude of the current but also its polarity. For instance, in the right panel of Fig. 2c, at $I = -2 \mu\text{A}$, the spin state of the magnetic domain is SAP, while it is AP state at $I = 2 \mu\text{A}$ determined from the red curve. In our revised paper, we have added the related discussion to rule out Joule heat as the cause of the unidirectional magnetic switching.

Comment #11: One minor point, please identify which electrode is grounded in Fig.4a/d.

Response: We appreciate the reviewer's valuable comment. The bottom (right in the figures) layer is grounded since its chemical potential is pinned at zero. This is consistent with Fig. 1a. To align our theoretical schematics with our experimental circuits, we have revised Fig. 4 and included this information in the figure caption.

Response to Referee #2:

Comment #1: This manuscript describes current-induced magnetization reversal (CIMR) in CrI₃ van der Waals heterostructure devices. CrI₃ is 2D magnet, which attracts much interest. The authors fabricated a device using double-layer CrI₃ with attaching graphene as the top and the bottom electrodes and demonstrated CIMR in the device. I show the following comments to the contents of the manuscript, which could contribute to evaluate the manuscript.

Response: We thank the reviewer for the positive perspective on the research topic by noting, "*CrI₃ is 2D magnet, which attracts much interest.*", and for providing the comments below to help us revise our manuscript. We have worked hard and carefully addressed the comments and anticipate his/her further consideration.

Comment #2: The concept of CIMR is well-established by many previous studies, and the authors' study does not show any significance and novelty in this perspective. In fact, novel physics of the CIMR in the CrI₃ devices is not included in this work. Hence, I found difficulty in finding any surprising and/or intriguing contents that warrants publication.

Response: We acknowledge the reviewer's comment that the concept of CIMR is well-established in traditional spintronics, evidenced by numerous prior studies. We regret any failure on our part to sufficiently highlight the innovative and significant aspects of our work, and we would like to clarify our perspectives on the reported results in this manuscript.

Firstly, investigations into CIMR in the field of 2D magnets are still rare, especially concerning how the tunneling current influences the spin states in unusual vdW magnetic domains of atomically thin 2D magnets. Our work represents the first study to address this question.

Secondly, different from traditional magnetic systems, 2D magnets like CrI₃ display distinctive interlayer exchange coupling. The existence of new mechanisms governing the tunneling current-controlled spin states in 2D magnets is still a subject of exploration. The manifestation of tunneling current-induced unidirectional spin state switching observed in few-layer CrI₃ reveals a new dimension in 2D magnet-based tunnel junctions. Given the anticipated collinear magnetizations of the CrI₃ layers, conventional spin transfer torque is inadequate to interpret our data directly. Our theoretical framework offers fresh insights into the mechanism underlying the observed unidirectional spin state transition.

Thirdly, the development of probabilistic bits leveraging metastable magnetic tunnel junctions constructed from conventional 3D magnets holds immense promise for advancing computing technologies, including probabilistic computing. Achieving controllable stochastic switching is the first step for realizing this ambition. Our work unveils the first instance of realizing and controlling such stochastic switching in 2D

atomically thin magnets through tunneling current. While immediate applications for creating a probabilistic bit are not ready, our findings pave the way for probing similar behaviors in other 2D magnets, even at room temperature and devising strategies to control their magnetism, encompassing metastable magnetic states.

To accentuate the significance and novelty of our study, we have made the corresponding modifications to our revised manuscript.

Comment #3: The weakness of the device is its high base resistance of the order of MegaOhm. Although the authors claimed energy efficient stochastic switching, this quite high base resistance strongly hinders applications. Thus, I do not agree with the authors' assertion that the device enables high density energy-efficient nanoscale logic gates and probabilistic/neuromorphic computing.

Response: We thank the reviewer for the comment regarding the magnitude of the tunneling resistance. We recognize that the tunneling resistance of magnetic tunnel junctions constructed from conventional 3D magnets can vary widely, ranging from tens of $k\Omega$ to a few $M\Omega$. In tunneling devices, the actual resistance is highly dependent on the amplitude of the bias current. For instance, in our device, near the observed unidirectional spin state transition depicted in Fig. 2, the measured voltage is approximately 0.6 V, and the applied current is around 3 μA , yielding a resistance of approximately 300 $k\Omega$. This is comparable to 3D magnetic tunnel junctions. Given that our device requires a significantly lower tunneling current to induce spin state transition, the energy needed to power the 2D magnet-based devices is indeed less than that for conventional magnetic tunnel junctions driven by STT. For instance, in comparison to conventional techniques referenced in Refs, 40 - 44 in the manuscript, our device manifests much lower power consumption. Specifically, traditional methods, such as CMOS-based ones, often consume substantially more power (e.g., CMOS-based: $\sim 2 \times 10^{-4}$ W, conventional metastable MTJ-based: $\sim 1 \times 10^{-5}$ W). Our device, with a tunneling current of 200 nA and a measured voltage of 0.45 V, has a power consumption of approximately 9×10^{-8} W, as depicted in Fig. 3a. This advantage may become even more pronounced after achieving similar effects in 2D magnets with a transition temperature near room temperature. Consequently, our research lays the groundwork for future studies of the role of tunneling current in controlling the magnetic state in other 2D magnets, thereby enabling the possibility of realizing high-density, energy-efficient spintronic and computing devices, such as nanoscale logic gates and probabilistic/neuromorphic computing. In the revised manuscript, we have revised our statement to highlight that our work unveils opportunities for designing and realizing such innovative applications. I hope this refined version aligns well with the reviewer's intended message.

Comment #4: Whereas the authors claimed the stochastic switching is significant, it is merely due to uncontrolled magnetic states (magnetic domains, total magnetization, coercive fields etc...), I think.

Response: Indeed, the stochastic switching observed in few-layer CrI₃ arises due to the metastable magnetic state, which is similar to that of metastable magnetic tunneling junction. The metastable states in our 2D magnets are highly controllable. We have demonstrated that the probability of this stochastic switching can be meticulously controlled by the tunneling current, as depicted in Fig. 3 and Extended Data Fig. 10. Additionally, we illustrated that the influence of magnetic field and temperature can further modulate the stochastic switching (refer to Extended Data Figs. 5-8). We believe that achieving controllable stochastic switching in 2D vdW magnets is significant. It not only furnishes a new platform and simpler device structure for designing future probabilistic bits but also has the potential to substantially mitigate power consumption.

Comment #5: Although the authors paid strong dedication to understand the underlying physics of the stochastic switching by combining with theory and experiments, missing and appearance of the symmetric switching at 1.5 K and 30 K was not sufficiently clear to me. Furthermore, if the authors' claim is current, the appearance of the switching can be in the positive bias in the other device because the net magnetization (magnetism) of two CrI₃ layers can be changed, and the probability of the appearance can be close to 50%. I also point out that the device characteristics are not fully controlled, and the authors implemented analyses only to a specific device, which hampers drawing and understanding of the whole picture of the observation if it is in this work.

Response: We are grateful to the reviewer for raising an important question regarding the expected occurrence of stochastic switching. At 1.5 K, it appears that stochastic switching is solely observed in the negative current region. However, at other temperatures (e.g., 30 K), stochastic switching is discernible in both positive and negative current regions. Indeed, similar to unidirectional spin state transition, the background magnetic field significantly influences the manifestation of stochastic switching. We have carefully checked our data and found out that when the temperature is 1.5 K, and the magnetic field is at 0.510T, stochastic switching can also be observed in the positive current region. We have incorporated the relevant zoomed-in view of the results in Extended Data Fig. 5 of the revised manuscript. In summary, we don't observe substantial differences between results measured at different temperatures when they are below the transition temperature. It's worth noting that the effects of the elevated temperature on stochastic switching include 1) the induction of thermal noise and 2) the reduction of the energy difference between SAP and SP states, resulting in a gradually reduced driving current.

Regarding the reviewer's comment on "...analyses only to a specific device," we wish to clarify that the experimental results, including 1) unidirectional spin state transition and 2) stochastic switching-are not confined to a specific device but were observed in devices with different thicknesses. Interestingly, we find that the number of layers serves as an additional degree of freedom in manipulating the stochastic switching.

For instance, we witness controllable stochastic switching between three spin states (Extended Data Fig. 10), which is directly associated with the unique layer-dependent magnetism in 2D CrI₃ magnets. Such control is unattainable in traditional magnetic tunnel junctions. To remove this confusion, we have added the related discussion in the revised manuscript.

Comment #6: In summary, I do not find sufficient significance and novelty that makes readers surprising in the manuscript at this stage. The authors tried to understand what takes place in their experiments. However, the device performance depends on the chance combination of the two CrI₃, which does not allow general understanding of profound and novel physics beyond the emblematic case of the authors' experiments.

Response: We thank all the reviewer's comments, which are valuable for us to improve our manuscript. In the response to the reviewer's comment # 5, we have clarified that the observed unidirectional spin state transition and stochastic switching are commonly observed in samples with different thicknesses and are not observed by luck. Our discovery has the potential to be demonstrated in other 2D magnets, enabling the opportunity of designing novel spintronic and computing devices. In the end, we have summarized the key findings of this work, including:

- 1) An unprecedented and deterministic tunneling current-induced spin-state transition between SA and SAP states in few-layer 2D magnets, with the switching direction governed by the bias polarity.
- 2) A novel insight into the tunneling current-induced magnetic switching in few-layer CrI₃, through the lens of nonequilibrium spin accumulation in graphene electrodes, which holds potential for broader applicability to other 2D magnets.
- 3) A pioneering demonstration of tunneling current-controlled stochastic switching between multiple metastable spin states within few-layer CrI₃, contingent on the number of layers.
- 4) Exceptional energy efficiency in CrI₃ tunnel junction devices, manifesting in power consumption at least three orders of magnitude lower than conventional magnetic tunnel junctions.

Reviewers' Comments:

Reviewer #1:

Remarks to the Author:

In the revised manuscript, the authors rephrased different paragraphs to clarify issues raised by the referees, such as add discussion of electrostatic gating, recalculate the current density in a more accurate way. These make the manuscript better, but do not clarify my concerns about the key findings of this work. The key information of this manuscript is the experimental finding of tunneling current induced unidirectional spin-state transition in CrI₃, and the theoretical interpretation involving nonequilibrium spin accumulation in graphene electrodes.

For the experimental side, I do not think current data are enough to support the claim. I can accept the definition of unidirectional spin-state transition, that is, depending on the polarity and amplitude of the current, as specified in the rebuttal letter "as the current amplitude increases at positive values, the switching transitions from SAP to SP. Conversely, the spin state switching reverses direction, from SP to SAP, within the region of negative current." This claim works well for demonstrated 2L device, but there is only one working 2L device. More important, it seems do not work for other devices. For 4L device shown in Extended Data Fig. 9a, the SAP to SP transition not only occur at positive current but also at negative current (from state 2 to state 1). For 5L device shown in Fig. S7, there is no observable spin state transition when current amplitude increases in both positive and negative current region.

For the theoretical interpretation side, I agree that the model in the manuscript indeed could explain the unidirectional spin-state transition and it is a nice one. But this model has quite some assumptions, even though they seems reasonable. I still think electric field and electrostatic doping effect can result in the observations in this manuscript, for example SP and SAP transition controlled by electric field has been demonstrated. As these effects have already been experimentally demonstrated, I suggest these effects should be discussed and excluded in manuscript so that nonequilibrium spin accumulation in graphene model can be better accepted.

Regarding above issue, the authors argued that there is not top or back gate in their CrI₃ tunnel junctions. But that is not necessary for build in an electric field in the device, for bias of 0.7V on 2L device, the electric field is around 0.5V/nm, which is quite large and can result in obvious effect as shown in previous reports (Nat. Mater. 17, 406–410 (2018); Nat. Nanotechnol. 13, 549–553 (2018); Nat. Nanotechnol. 13, 544–548 (2018)). The authors already observed strong voltage dependence of switching field in extended data Fig.3 and Fig.4, similar as previous reports, being another evidence that electric field/doping can have effects in studied device. Beside the argument of no top or back gate, the authors also argued that unidirectional magnetism state switching was observed in this work, different from previous report. This difference can be explained that there are different type of domains manipulated in the device, type "down/up" and type "up/down" that is manipulated by positive and negative electric field respectively. The assumption that more than one domain are involved can be evidenced by Fig.2c, for red curve, it is hard to understand one domain changed from SP (2uA) to SAP (1uA), back to SP (-1uA) then to SAP (-2uA), more reasonable picture is that the SAP state at 1uA and SP state at -1uA are for different domains.

If the authors can address above two issues about experimental identification of unidirectional spin-state transition and interpretation with electric field/doping effect, I would be very happy to recommend the publication of revised manuscript in Nature Communications.

Reviewer #2:

Remarks to the Author:

I thank the authors for their dedication to respond to my comments in the previous round. I have mainly raised some concerns about the novelty of the authors' study, which hinders recommending the manuscript for publication in Nature Communications.

Although the authors claimed that "...investigation into CIMR in the field of 2D magnets are still rare, especially concerning how the tunneling current influences the spin states...", I do not find

novelty in magnetism and spintronics in the conclusions of the study. I also would like to point out that existence of the interlayer exchange coupling in layered ferromagnets is trivial and that I do not find any novel physics related with magnetism in the study. Indeed, the authors did not clearly show what is the novel physics that has not been found in magnetism.

Regarding the p-bit operation, the current status of the related research field is targeting actual computation. Meanwhile, the authors have not demonstrated any computation using their devices in this work. Related comment is on very small on/off voltage differences in the p-bit operation. The difference is merely 10 mV, whereas it was 5 V in MTH p-bits. Although the operating power is small as the authors claimed (but I still express concern that the minimum base resistance of 300 kOhm in the authors' device is still three orders of magnitude greater than that of a conventional MTJ (please see APL 122, 112404 (2023), and the authors' claim, "This is comparable to 3D magnetic tunnel junctions" is not correct) the quite small on/off difference strongly impedes reliable and efficient computation using p-bit systems. To be honest, I do not think the same level of computation in the previous studies cannot be realized by using the authors' p-bit devices. To rebut my criticism, the authors are requested to carry out actual computation as Borders et al. realized.

Regarding the finding of novel physics, the underline physics of the results the authors obtained is still murky and clear description is not found in the text. In fact, the authors repeated the term "novel physics", "novel insight"... but it is still quite unclear to me what is the novel physics/insight that can surpass/transcend.challenge conventional understandings in magnetism, spintronics and related fields.

Response to referee #1:

Comment #1: In the revised manuscript, the authors rephrased different paragraphs to clarify issues raised by the referees, such as add discussion of electrostatic gating, recalculate the current density in a more accurate way. These make the manuscript better, but do not clarify my concerns about the key findings of this work. The key information of this manuscript is the experimental finding of tunneling current induced unidirectional spin-state transition in CrI₃, and the theoretical interpretation involving nonequilibrium spin accumulation in graphene electrodes.

Response: We are grateful for the referee's recognition of our past efforts in addressing the concerns raised by the referees and improving the manuscript accordingly, notably encapsulated in the comment "*These make the manuscript better*". We acknowledge that certain issues (detailed below) raised by the referee may not have been satisfactorily resolved in our previous responses and revisions. In this current revision, we have carefully considered all the comments/concerns and have made corresponding changes to our manuscript. We believe that these revisions appropriately address the concerns raised, as reflected in our current response and the updated version of the manuscript.

Comment #2: For the experimental side, I do not think current data are enough to support the claim. I can accept the definition of unidirectional spin-state transition, that is, depending on the polarity and amplitude of the current, as specified in the rebuttal letter "as the current amplitude increases at positive values, the switching transitions from SAP to SP. Conversely, the spin state switching reverses direction, from SP to SAP, within the region of negative current." This claim works well for demonstrated 2L device, but there is only one working 2L device. More important, it seems do not work for other devices. For 4L device shown in Extended Data Fig. 9a, the SAP to SP transition not only occurs at positive current but also at negative current (from state 2 to state 1). For 5L device shown in Fig. S7, there is no observable spin state transition when current amplitude increases in both positive and negative current region.

Response: We thank the referee's feedback and the comment "*This claim works well for demonstrated 2L device*". We agree that validating the unidirectional switching behavior in another 2L CrI₃ device is crucial for our study. To address this, we have fabricated a new 2L CrI₃ device and have successfully demonstrated tunneling current-induced unidirectional spin-state switching in the low bias current region, as depicted in Fig. R1. Our results show that as the amplitude of the tunneling current increases, the transition from SP to SAP occurs at approximately -70 nA, while the SAP to SP transition is noted at around 20 nA. These findings have been incorporated into our revised supplementary information, where we have updated Table S1 accordingly.

We concur with the referee’s observation that in the 4L sample, the SAP (state 2) to SP (state 1) transition occurs near -500 nA, but it is in the region where stochastic switching is evident, as shown in Extended Data Fig. 9a. We note that a similar phenomenon is observed in the 2L CrI₃ at about -4.5 μ A, as indicated in Fig. 1a. To clarify, the unidirectional spin-state switching is primarily applicable in the region of relatively low tunneling currents. This means the unidirectional spin-state transition is expected to occur before the onset of stochastic switching, a point at which the spins from different layers become non-collinear due to current-induced thermal noise. Consequently, as discussed in the manuscript, our theoretical model is not applicable in the relatively high-current region. Nonetheless, within the parameters of this condition, the unidirectional spin-state transition induced by tunneling currents is still observable in the 4L CrI₃ device. Specifically, the transition from SAP (state 4) to SP (state 5) is noted at approximately 200 nA, and the reverse transition from SP (state 3) to SAP (state 2) occurs around -200 nA, as illustrated in Extended Data Fig. 9a. To further alleviate the possible confusion, we have revised our manuscript accordingly by emphasizing that the tunneling current-induced unidirectional spin-state transition occurs in the relatively low bias current region.

In the case of the 5L CrI₃ device, unidirectional spin-state switching is also evident from the left panel of Fig. S7 at $B = 0.61$ T. For example, the transition from SP to SAP is observed at approximately 750 nA, while the transition from SAP to SP occurs at around -1500 nA. We note that the SAP to SP transition occurs in the negative current region for this specific sample, which is a deviation from the behavior observed in other samples. Our theoretical model can readily explain this variation, which posits that the magnetic domain in the upper layer has a greater magnetization. In contrast, only stochastic switching is observed in the right panel of Fig. S7 at $B = 0.586$ T. As discussed in the manuscript (see Extended Data Figs. 5-8), the magnetic field strongly affects the unidirectional spin-state switching. At this particular magnetic field, the system is moved out of the spin-state transition region, which explains the absence of unidirectional switching and the presence of only stochastic switching.

Fig. R1. Transport properties of another 2L CrI₃ tunneling device. (a) Tunneling resistance as a function of the magnetic field measured at $T = 1.5$ K. (b) Corresponding voltage-current characteristics measured at $T = 1.5$ K and $B = 0.57$ T.

Comment # 3: For the theoretical interpretation side, I agree that the model in the manuscript indeed could explain the unidirectional spin-state transition and it is a nice one. But this model has quite some assumptions, even though they seem reasonable. I still think electric field and electrostatic doping effect can result in the observations in this manuscript, for example SP and SAP transition controlled by electric field has been demonstrated. As these effects have already been experimentally demonstrated, I suggest these effects should be discussed and excluded in manuscript so that nonequilibrium spin accumulation in graphene model can be better accepted.

Regarding above issue, the authors argued that there is not top or back gate in their CrI₃ tunnel junctions. But that is not necessary for build in an electric field in the device, for bias of 0.7V on 2L device, the electric field is around 0.5V/nm, which is quite large and can result in obvious effect as shown in previous reports (Nat. Mater. 17, 406–410 (2018); Nat. Nanotechnol. 13, 549–553 (2018); Nat. Nanotechnol. 13, 544–548 (2018)). The authors already observed strong voltage dependence of switching field in extended data Fig.3 and Fig.4, similar as previous reports, being another evidence that electric field/doping can have effects in studied device. Beside the argument of no top or back gate, the authors also argued that unidirectional magnetism state switching was observed in this work, different from previous report. This difference can be explained that there are different type of domains manipulated in the device, type “down/up” and type “up/down” that is manipulated by positive and negative electric field respectively. The assumption that more than one domain are involved can be evidenced by Fig.2c, for red curve, it is hard to understand one domain changed from SP (2uA) to SAP (1uA), back to SP (-1uA) then to SAP (-2uA), more reasonable picture is that the SAP state at 1uA and SP state at -1uA are for different domains.

Response: We acknowledge and appreciate the referee’s concerns regarding the potential effects of electric fields and electrostatic doping on our experimental results that were not sufficiently considered in our original and previously revised manuscripts. We recognize that electric field and electrostatic doping can influence the magnetism of magnetic materials differently. It is, therefore, essential to discern which factor - electric field or electrostatic doping - predominantly affects the manipulation of spin states in few-layer CrI₃. This consideration is crucial for a comprehensive understanding of our findings.

We agree with the referee’s comment that gate-voltage-induced spin-state switching between SP and SAP states has been previously demonstrated, as seen in notable publications like Nat. Mater. 17, 406–410 (2018); Nat. Nanotechnol. 13, 549–553 (2018); and Nat. Nanotechnol. 13, 544–548 (2018). It’s important to note that electrical fields were applied using top and/or bottom gates in these reports. In devices with

such gate configurations, distinguishing the effects of electric field and electrostatic doping on spin-state switching between SP and SA in few-layer CrI₃ is not straightforward. However, the authors of Nat. Nanotechnol. 13, 549–553 (2018) managed to overcome this challenge by employing a specialized circuit design. They conclusively demonstrated that the layer-related spin-state switching in few-layer CrI₃ is predominantly induced by electrostatic doping rather than the electric field, as elaborated in their main text and Supplementary Section 3. Our devices generate electric fields through two graphene electrodes, not the top and/or bottom gates. While the magnitude of these electric fields is similar to those generated by gate voltages in the aforementioned studies, the electrostatic doping effect in our device is minimal. Otherwise, we should be able to see the layer-magnetization switching in our devices as in the three reports cited above. Therefore, our work also demonstrates that the pure electric field cannot induce layer-magnetization switching in few-layer CrI₃. Consequently, the effect of the electric field on unidirectional spin-state switching in our devices would be considerably weak.

Regarding the comment “...there are different type of domains manipulated in the device, type “down/up” and type “up/down” that is manipulated by positive and negative electric field respectively...” and “The assumption that more than one domain are involved can be evidenced by Fig.2c, for red curve, it is hard to understand one domain changed from SP (2uA) to SAP (1uA), back to SP (-1uA) then to SAP (-2uA), more reasonable picture is that the SAP state at 1uA and SP state at -1uA are for different domains” We agree with the referee’s observation regarding the presence of various types of magnetic domains in our system. This is in line with our transport data, which indicates multiple magnetic domains in a few-layer CrI₃. For instance, in Figs. 2a-c and Extended Data Figs. 5-10, we frequently observe multiple step-like voltage changes within single I-V curves. These changes suggest tunneling current-induced spin-state transitions occurring across multiple magnetic domains. Moreover, considering the distinct coercivities of different magnetic domains, the hysteresis behavior, and the bias voltage (bias current) dependent coercivities of magnetic domains (Extended Data Figs. 3 and 4), it seems improbable that the transitions observed from SP (2uA) to SAP (1uA), then back to SP (-1uA), and subsequently to SAP (-2uA) originate from the same magnetic domain.

The hypothesis that “This difference can be explained that there are different type of domains manipulated in the device, type “down/up” and type “up/down” that is manipulated by positive and negative electric field respectively” might not satisfactorily account for the unidirectional spin-state transition we observed. Firstly, the effect of an electric field on magnetism in our device structures appears to be minimal. Secondly, even if the electric field could strongly affect the spin-states in the CrI₃ devices, the unidirectional spin-state transition is not replicated. For instance, in Fig. 4a of the paper published in Nat. Mater. 17, 406–410 (2018), a positive electric field sustains the SAP (either “down/up” or “up/down”)

state, contrasting with the SP ('down/down' or 'up/up') state at negative electric fields. Therefore, observing a transition from the SAP to SP state in the region of positive bias current, as we have observed, appears unlikely to be induced by positive electric fields. Similarly, the transition from SP to SAP state, which we achieved using negative tunneling current, seems improbable to be realized by negative electric fields. Thirdly, electrostatic doping typically induces gradual spin-state transitions, in contrast to our experiments' sharp, step-like transitions.

In our revised manuscript, we have added a paragraph at the end of the theoretical model section to discuss the potential influence of electric fields on the unidirectional spin-state transitions observed in our CrI₃ devices.

Comment # 4: If the authors can address above two issues about experimental identification of unidirectional spin-state transition and interpretation with electric field/doping effect, I would be very happy to recommend the publication of revised manuscript in Nature Communications.

Response: We are grateful for the insightful comments and suggestions provided by the referee. We have carefully considered each point raised and believe we have addressed them thoroughly in our responses and the revised manuscript. We look forward to the referee's further evaluation and feedback.

Response to referee #2:

Comment #1: I thank the authors for their dedication to respond to my comments in the previous round. I have mainly raised some concerns about the novelty of the authors' study, which hinders recommending the manuscript for publication in Nature Communications.

Response: We appreciate the time and effort the referee has dedicated to evaluating our revised manuscript and responses. We have diligently addressed the concerns and comments raised and further emphasized the novelty of our work in this revision round. We look forward to the referee's further evaluation and feedback.

Comment #2: Although the authors claimed that "...investigation into CIMR in the field of 2D magnets are still rare, especially concerning how the tunneling current influences the spin states...", I do not find novelty in magnetism and spintronics in the conclusions of the study. I also would like to point out that existence of the interlayer exchange coupling in layered ferromagnets is trivial and that I do not find any novel physics related with magnetism in the study. Indeed, the authors did not clearly show what is the novel physics that has not been found in magnetism.

Response: We thank the referee for the feedback. Our work gains context from the foundational discoveries in 2017, where two independent studies (*Nature* 546, 265 (2017) and *Nature* 546, 270 (2017)) demonstrated the existence of stable magnetism in 2D vdW magnets at the 2D limit. Since then, a series of advancements (such as *Nat. Mater.* 17, 406–410 (2018); *Nat. Nanotechnol.* 13, 549–553 (2018); *Nat. Nanotechnol.* 13, 544–548 (2018); *Nat. Mater.* 18, 1298–1302 (2019); *Nat. Mater.* 18, 1303–1308 (2019); *Nature Materials* 21, 1373 (2022), etc.) have propelled the study of 2D magnetism in 2D vdW magnets to the forefront of condensed matter physics and spintronics research. However, a gap remains in exploring layer (global) spin states in few-layer 2D vdW magnets, particularly regarding manipulating spin states within magnetic domains. Our work addresses this gap through both experimental and theoretical demonstrations. We have achieved a unidirectional spin-state transition and have experimentally demonstrated tunneling current controllable stochastic switching among magnetic domains in few-layer CrI₃. Our work represents a significant leap forward in studying 2D magnetism in 2D vdW magnets. We believe our contributions significantly advance the understanding of 2D magnetism and open new avenues for the potential applications of 2D vdW magnets in advanced computing. The novelty and significance of our work have been clarified and underscored both in our revised manuscript and our detailed point-by-point responses. Below are the key highlights of our work:

- (1) We have achieved an unprecedented tunneling current-induced unidirectional spin-state switching between SP and SAP states in few-layer 2D magnets. This switching is uniquely characterized by its

dependence on both the magnitude and the bias polarity of the tunneling current, a novel finding in the field.

- (2) Our discovery opens up a new avenue of exploration into the mechanisms governing tunneling current-controlled spin states in 2D magnets. The observed unidirectional spin-state switching in few-layer CrI₃, especially considering the expected collinear magnetizations of its layers, cannot be explained by conventional spin transfer torque theory. Our theoretical model provides new insights into this phenomenon, suggesting broader applicability to other 2D magnetic materials.
- (3) We present a pioneering demonstration of tunneling current-controlled stochastic switching between multiple metastable spin states within magnetic domains of few-layer CrI₃. This phenomenon is notably influenced by the number of layers in the material, marking a significant advancement in pushing the potential applications of 2D magnetic systems.
- (4) Our work introduces a new method for manipulating metastable spin states in magnetic domains of few-layer CrI₃ through tunneling currents. This represents a significant step forward in the study of 2D vdW magnets and is poised to stimulate further experimental and theoretical research, offering new possibilities for the development of future computing technologies.

Comment #3: Regarding the p-bit operation, the current status of the related research field is targeting actual computation. Meanwhile, the authors have not demonstrated any computation using their devices in this work. Related comment is on very small on/off voltage differences in the p-bit operation. The difference is merely 10 mV, whereas it was 5 V in MTH p-bits. Although the operating power is small as the authors claimed (but I still express concern that the minimum base resistance of 300 kOhm in the authors' device is still three orders of magnitude greater than that of a conventional MTJ (please see APL 122, 112404 (2023), and the authors' claim, "This is comparable to 3D magnetic tunnel junctions" is not correct) the quite small on/off difference strongly impedes reliable and efficient computation using p-bit systems. To be honest, I do not think the same level of computation in the previous studies cannot be realized by using the authors' p-bit devices. To rebut my criticism, the authors are requested to carry out actual computation as Borders et al. realized.

Response: We are grateful for the referee's enthusiasm and suggestion to demonstrate actual computation using 2D vdW magnet-based probabilistic bits (p-bits). This aligns closely with the trajectory of our future research efforts. We wish to highlight that our current work represents a pioneering proof of concept in realizing stochastic switching in few-layer 2D magnets, an essential step toward practical probabilistic computing. We acknowledge that advancing from this proof of concept to actual probabilistic computing devices will require significant effort. Critical areas for improvement include enhancing the on/off voltage difference, optimizing the junction resistance, exploring different material

systems, refining the design of the p-bits, etc. However, these challenges present exciting opportunities for future research. Encouragingly, 2D magnetism has already been achieved near room temperature, and a variety of air-stable 2D vdW magnets have been identified. The stochastic switching we have demonstrated is not unique to few-layer CrI₃. Therefore, we are optimistic that more suitable 2D magnetic systems exist, which could offer improved performance for real probabilistic computing applications. The significance of our work lies in establishing a foundational understanding and demonstrating the feasibility of stochastic switching in 2D magnets. This lays the groundwork for future studies to build upon, potentially leading to breakthroughs in probabilistic computing. We are excited about the possibilities that our work opens up and look forward to contributing further to this rapidly evolving field.

Comment #4: Regarding the finding of novel physics, the underline physics of the results the authors obtained is still murky and clear description is not found in the text. In fact, the authors repeated the term “novel physics”, “novel insight”..., but it is still quite unclear to me what is the novel physics/insight that can surpass/transcend.challenge conventional understandings in magnetism, spintronics and related fields.

Response: We appreciate the opportunity to respond to the referee's comments and have endeavored to clearly articulate the novelty and significance of our experimental and theoretical work. We regret that our previous responses and manuscript revisions did not fully convey the intended message to the referee. In light of this, we have undertaken additional revisions to our manuscript, aiming to highlight our research's key aspects and innovations more effectively. We believe these latest revisions offer a clearer understanding of our work's contributions to the field. We sincerely look forward to the referee's further evaluation and are open to any additional feedback or suggestions that might help enhance our study's clarity and impact.

Reviewers' Comments:

Reviewer #1:

Remarks to the Author:

I appreciate authors' effort of fabricating extra bilayer device and additional discussions to exclude the possible origin of electric field/doping effect, these mainly addressed my concerns about this manuscript, thus I support the publication on Nature Communications.